# Structures of *Saccharolobus solfataricus* initiation complexes with leaderless mRNAs highlight archaeal features and eukaryotic proximity

Gabrielle Bourgeois[1,5], Pierre-Damien Coureux[1,4,5], Christine Lazennec-Schurdevin[1], Clément Madru [1], Thomas Gaillard [1], Magalie Duchateau [2], Julia Chamot-Rooke [2], Sophie Bourcier[3], Yves Mechulam [1] & Emmanuelle Schmitt [1] ✉

The archaeal ribosome is of the eukaryotic type. TACK and Asgard superphyla, the closest relatives of eukaryotes, have ribosomes containing eukaryotic ribosomal proteins not found in other archaea, eS25, eS26 and eS30. Here, we investigate the case of *Saccharolobus solfataricus*, a TACK crenarchaeon, using mainly leaderless mRNAs. We characterize the small ribosomal subunit of *S. solfataricus* bound to SD-leadered or leaderless mRNAs. Cryo-EM structures show eS25, eS26 and eS30 bound to the small subunit. We identify two ribosomal proteins, aS33 and aS34, and an additional domain of eS6. Leaderless mRNAs are bound to the small subunit with contribution of their 5′-triphosphate group. Archaeal eS26 binds to the mRNA exit channel wrapped around the 3′ end of rRNA, as in eukaryotes. Its position is not compatible with an SD:antiSD duplex. Our results suggest a positive role of eS26 in leaderless mRNAs translation and possible evolutionary routes from archaeal to eukaryotic translation.

Archaea are widespread on earth, living not only in extreme environments but also in very common biotopes such as soil or the human microbiota[1–4]. Considering fundamental genetic information processing mechanisms, such as protein biosynthesis, archaea are close to eukaryotes. Recent phylogenetic models favor a two-domain tree of life, bacteria and archaea, with eukaryotes that could have emerged from within an archaeal branch. Within the tree of life, the archaeal domain itself is rapidly being enriched with new branches thanks to extensive identification of new species, showing an enormous diversity whose study will contribute to better understanding of evolutionary relationships with bacteria and eukaryotes[5–15].

The archaeal ribosome is of the eukaryotic type. It contains ribosomal proteins that are either universal or specific to the eukaryotic and archaeal domains[16–25]. Interestingly, composition of the archaeal ribosome varies according to archaeal phyla. Diversity of archaeal ribosomes was first observed by comparing halobacterial and crenarchaeotal ribosomes with bacterial and eukaryotic ribosomes using electron microscopy[26,27]. Ribosomes from Crenarchaeotes (at that time called Eocytes and also called now Thermoprotei[15,28]) were

[1]Laboratoire de Biologie Structurale de la Cellule (BIOC), CNRS, Ecole polytechnique, Institut Polytechnique de Paris, Palaiseau 91120, France. [2]Institut Pasteur, Université Paris Cité, CNRS UAR 2024, Mass Spectrometry for Biology, Paris 75015, France. [3]Laboratoire de Chimie Moléculaire (LCM), CNRS, Ecole polytechnique, Institut Polytechnique de Paris, Palaiseau 91120, France. [4]Present address: Retroviruses and Structural Biochemistry Team, Molecular Microbiology and Structural Biochemistry, UMR 5086 CNRS-Lyon 1, CNRS, Université de Lyon, Lyon, France. [5]These authors contributed equally: Gabrielle Bourgeois, Pierre-Damien Coureux. ✉e-mail: emmanuelle.schmitt@polytechnique.edu

found closer to the eukaryotic ones leading to the idea that eukaryotes may have originated from within an archaeal phylum instead of being a sister lineage[29,30]. Later, the TACK superphylum, including Thaumarchaeota, Aigarchaeota, Crenarchaeota and Korarchaeota, was defined. Interestingly, TACK archaea have orthologs of eukaryotic ribosomal proteins not found in other archaeal phyla[31-33] such as small subunit proteins eS25, eS26 and eS30. More recently, Bathyarchaeota[34] and Verstraeteachaeota[35] were added to TACK (also defined as Thermoproteota in the Genome Taxonomy Database[15]). Finally, it was also discovered that Asgard (Loki-, Odin-, Thor-, Heimdall-, Hel-) archaeota also shared these ribosomal proteins with TACK[8,36]. Studies in eukaryotes showed that eS25, eS26 and eS30 are involved in translation initiation[37-39]. However no insight into their function in archaea has been obtained yet.

Archaeal translation initiation has both bacterial and eukaryotic features[40-43]. Archaeal initiation factors, aIF1, aIF1A, aIF2, and aIF5B correspond to a subset of eukaryotic translation initiation factors. In contrast, archaeal mRNAs are not processed after transcription and are therefore of the bacterial type. Transcriptomic analyzes (e.g[44-53]) have shown that depending on the archaeal phylum, mRNAs mainly contain Shine-Dalgarno (SD) sequences or have very short ($\leq$ 5 nucleotides) or no 5′-untranslated regions (UTR). mRNAs belonging to the latter class are called leaderless. The whole data suggest that leaderless mRNAs and leadered mRNAs with SD sequences co-exist in all archaea. However, in some archaea, most mRNAs are leaderless, for instance the euryarchaeon *Haloferax volcanii* (72%)[51] or the crenarchaeon *Saccharolobus solfataricus* (73 %)[48].

Up to now, only few examples of archaeal ribosome structures are known[54-63]. Moreover, most of them come from studies of Thermococcales, belonging to the Euryarchaeota superphylum, distant from TACK and Asgard. Recently, a cryo-EM study of the translocation mechanism in a crenarchaeal ribosome, that of *Sulfolobus acidocaldarius*, has been published[63]. However, the resolution of the cryo-EM maps was not sufficient to identify and describe the crenarchaeotal small subunit specificities. Moreover, the proteins eS25, eS26 and eS30 were not modeled.

Here, we study 30S translation initiation complexes from *S. solfataricus*[64] prepared with mRNAs having different 5′ untranslated regions. Biochemical characterization and cryo-EM structures identify unexpected specificities of the small crenarchaeotal subunit that illustrate the evolutionary diversity in archaea. Moreover, our work shows the archaeal versions of the ribosomal proteins eS25, eS26 and eS30. Sequence specificities of eS25 and eS30 are discussed at the light of their role in the translation. We also study leaderless mRNAs binding and show the contribution of the 5′-triphosphate group. Finally, we show that the binding of eS26 in the mRNA exit channel is not compatible with the SD:antiSD duplex. This suggests that eS26 could regulate leaderless versus leadered mRNA translations. Collectively, our work is part of the current trend towards comparative studies (e.g[65-70]) that provide a better understanding of the evolution of the translation machinery.

## Results

### Reconstitution of S. solfataricus initiation complexes

The importance of polyamines and low salt concentration in promoting activity of crenarchaeal ribosomes was previously observed[71-74]. In line with these pioneering studies, we noted that when *S. solfataricus* 30S subunits were prepared in a buffer containing 100 mM NH$_4$Cl (20 mM MOPS pH 6.7, 100 mM NH$_4$Cl, 10.5 mM Mg acetate, 0.1 mM EDTA, 6 mM 2-mercaptoethanol), we were unable to detect toeprinting signals for an initiation complex composed of an SD-leadered mRNA, Met-tRNA$_i^{Met}$ and aIF2. In contrast, when the ribosomes were purified in a buffer containing low NH$_4$Cl, high Mg acetate and 2.5 mM spermine, toeprinting signals were observed. Therefore, we used buffer A (20 mM MOPS pH 6.7, 10 mM NH$_4$Cl, 18 mM Mg acetate,

2.5 mM Spermine, 0.1 mM EDTA, 6 mM 2-mercaptoethanol) in all our further ribosomal complex preparations and biochemical experiments. Quality of the sucrose gradient-purified 30S and 50S was analyzed by SDS-PAGE (Supplementary fig. 1a and b). Moreover, bottom-up proteomics was performed on purified 30S to check for the presence of the ribosomal proteins predicted from genome analysis (Supplementary fig. 2, Methods). To ensure the quality of our ribosomal preparations, we also measured the activity of the purified ribosomes using standard poly(U)-directed in vitro translation assay for poly(Phe) synthesis as described previously[71-73] (Methods, Supplementary fig. 1c). Overall, the data indicated that our ribosomal subunit preparations were active and amenable to further structural analysis.

We then used toeprinting assays to test the stability of reconstituted translation initiation complexes assembled on leadered or leaderless mRNAs with or without aIF2 and Met-tRNA$_i^{Met}$. Three mRNAs were chosen from the transcriptomic data[48]. Two of them are leaderless mRNAs starting with an AUG (Ss-MAP) or a GUG (Ss-aIF2β) start codon. One mRNA has a 5′UTR region containing a 5-base SD motif (Ss-aEF1A-like). These mRNAs were produced by in vitro transcription. Ss-aIF2β lmRNA and Ss-aEF1A-like have a 5′-triphosphate extremity whereas Ss-MAP, produced using a hammerhead ribozyme construct, has a 5′-OH extremity (Methods, Supplementary Table 1). These mRNAs were used in toeprinting experiments (Supplementary fig. 3). The toeprinting signals were compared to those obtained with a model mRNA containing a 9-base SD sequence, derived from the natural *P. abyssi* aEF1A mRNA (model-SD mRNA)[57,75]. As shown in Fig. 1a and in Supplementary fig. 3, we observed toeprinting signals with all mRNAs when 30S:mRNA complexes were formed in the presence of Met-tRNA$_i^{Met}$ and aIF2, with or without aIF1A. Faint arrests were also detected with 30S:mRNA complexes with both leadered and leaderless mRNAs. However, as already observed for *Pyrococcus abyssi* initiation complexes[75], in the presence of the ternary complex, aIF2:GDPNP:Met-tRNA$_i^{Met}$, the toeprinting signal increased. This is consistent with the tight binding of aIF2 to Met-tRNA$_i^{Met}$[76,77]. No obvious effect of aIF1A on the toeprinting signal intensity was observed. Overall, these results indicate that 30S initiation complexes with leaderless mRNAs can be reconstituted in vitro. RT arrests were observed with both leaderless mRNAs. However, with Ss-MAP lmRNA that was prepared with a 5′-OH extremity, the concentration of tRNA and factors had to be increased to obtain significant toeprinting signal (Fig. 1a and Supplementary fig. 3).

To compare the molecular basis of mRNA translation with or without 5′ leader, we prepared 30S:mRNA:aIF2:Met-tRNA$_i^{Met}$:aIF1A complexes using short versions of the four mRNAs described above (Supplementary Table 1, Methods). SD-leadered mRNAs and Ss-MAP lmRNA were chemically synthetized, with Ss-MAP lmRNA having a 5′-triphosphate extremity (Eurogentec). A 15-nucleotide version of Ss-aIF2β lmRNA was produced by in vitro transcription, and thus contained a 5′ triphosphate (see "Methods"). The initiation complex (IC) containing the model-SD mRNA was purified by size exclusion chromatography whereas the three other ICs were purified by affinity chromatography using His-tagged versions of the α and β subunits of aIF2 (Supplementary fig. 4). Cryo-EM datasets were collected for each IC. Nomenclature of the datasets is described in Figs. 2, Supplementary fig. 5 and Supplementary Table 2. DS1 to DS3 were collected on Titan Krios 300 kV microscopes and DS4 was collected on a Glacios 200 kV microscope. After 2D classifications and a first round of 3D classification, refinement with the full particle sets showed some heterogeneity at the level of the conformation of the distal part of helix h44. Therefore, the datasets were classified either using a mask on the SSU body (DS1 and DS2) or using head subtraction (DS3) to sort helix h44 conformations and the presence/absence of the initiator tRNA. From these 3D classifications, we isolated two 30S conformations with Met-tRNA$_i^{Met}$ bound to the P site (Supplementary figs. 5 and 6). One conformation showed helix h44 positioned as expected on the body of the

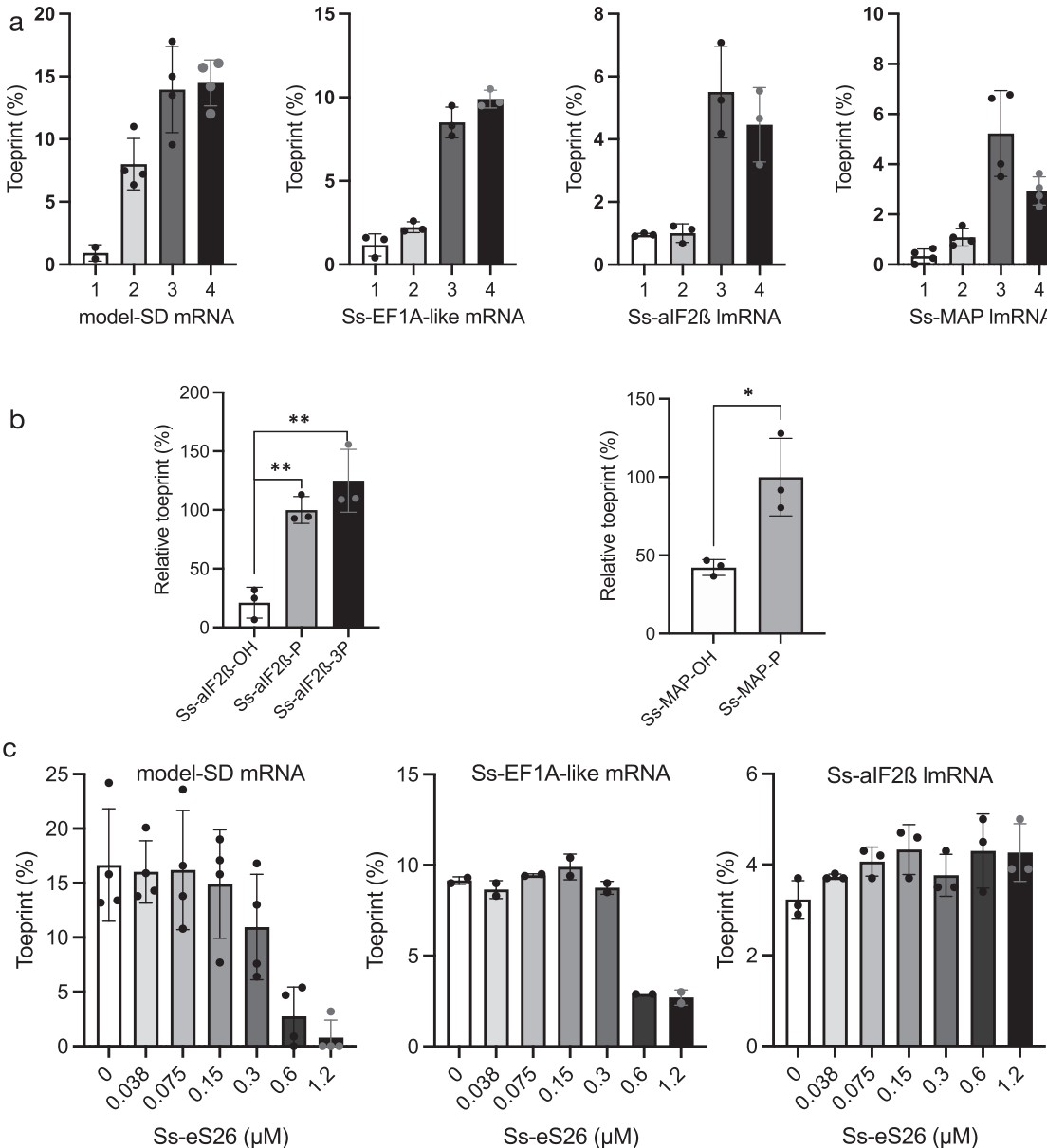

**Fig. 1 | Toeprinting signal of 30S complexes assembled on various mRNAs.**
**a** Toeprint signals for (1) 30S:mRNA, (2) 30S:mRNA:Met-tRNA$_i^{Met}$, (3) 30S:mRNA:aIF2:Met-tRNA$_i^{Met}$ (4) 30S:mRNA:aIF2:Met-tRNA$_i^{Met}$:aIF1A were measured as described in Methods. Data are presented as means ± SD from independent experimental units (Methods). Dots show individual data points. Typical experiments and the sequences of the mRNAs are shown in Supplementary fig. 3 and Supplementary Table 1. Ss-MAP mRNA has a 5'OH extremity whereas the Ss-aIF2β mRNA has a 5' triphosphate group. Experimental toeprinting conditions were the same for model-SD mRNA ($n = 4$), Ss-EF1A-like mRNA ($n = 3$) and Ss-aIF2β lmRNA ($n = 3$). With Ss-MAP lmRNA ($n = 4$), the molar excesses of IF and tRNA with respect to 30S were increased by factors of 4 and 2.5, respectively. **b** Importance of the 5'-triphosphate end. Relative toeprinting signals were measured for 5'-triphosphate (Ss-aIF2β only), 5' monophosphate or 5'-OH versions of the Ss-aIF2β and Ss-MAP lmRNAs. The values represented are the means and standard deviations from 3 independent experiments. For each mRNA, values were normalized to the mean obtained with the monophosphorylated mRNA which was given the arbitrary value of 100%. Two-tailed *P* values were calculated from unpaired t tests in PRISM (*P* values were 0.0014 and 0.0038 for Ss-aIF2β and 0.017 for Ss-MAP). Typical experiments are shown in Supplementary fig. 17. **c** Influence of Ss-eS26 concentration on the main toeprinting signal intensity obtained with 30S:mRNA:aIF2: Met-tRNA$_i^{Met}$ complexes. Ss-eS26 concentrations ranged from 0 to 1.2 μM. 30S concentration in all experiments was 0.1 μM (Supplementary fig. 20). The means from independent experiments (left: $n = 4$; middle: $n = 2$; right: $n = 3$) with the calculated standard deviations are represented. Dots show individual data points. Source data are provided as a Source Data file.

30S, hereafter named h44-down (Supplementary fig. 6a). The other conformation showed a position of the h44 helix that deviated from h44-down from the C1373-G1448 base pair. Helix h44 is detached from the body and becomes fuzzy after the G1381•A1440 base pair. This second h44 helix position was hereafter named h44-up (Supplementary fig. 6b). The upper part of helix h44 is connected to helix h45 and helix h24 and forms A and P sites identically in the h44-up and -down structures (Supplementary fig. 6). The modified bases located around the active sites are also visible in the two conformations. Accordingly, the initiator tRNA and the mRNA are bound similarly in the h44-up and in the h44-down structures (Supplementary fig. 6). Finally, the comparison of the IC structures with h44-up or -down did not reveal any changes beyond the position of the distal part of helix h44 with respect to the body of the SSU. Therefore, the various positions of helix h44 observed here do not reflect and do not resemble intermediate biogenesis states, as observed in human[78,79] or mitochondrial SSU[80], that

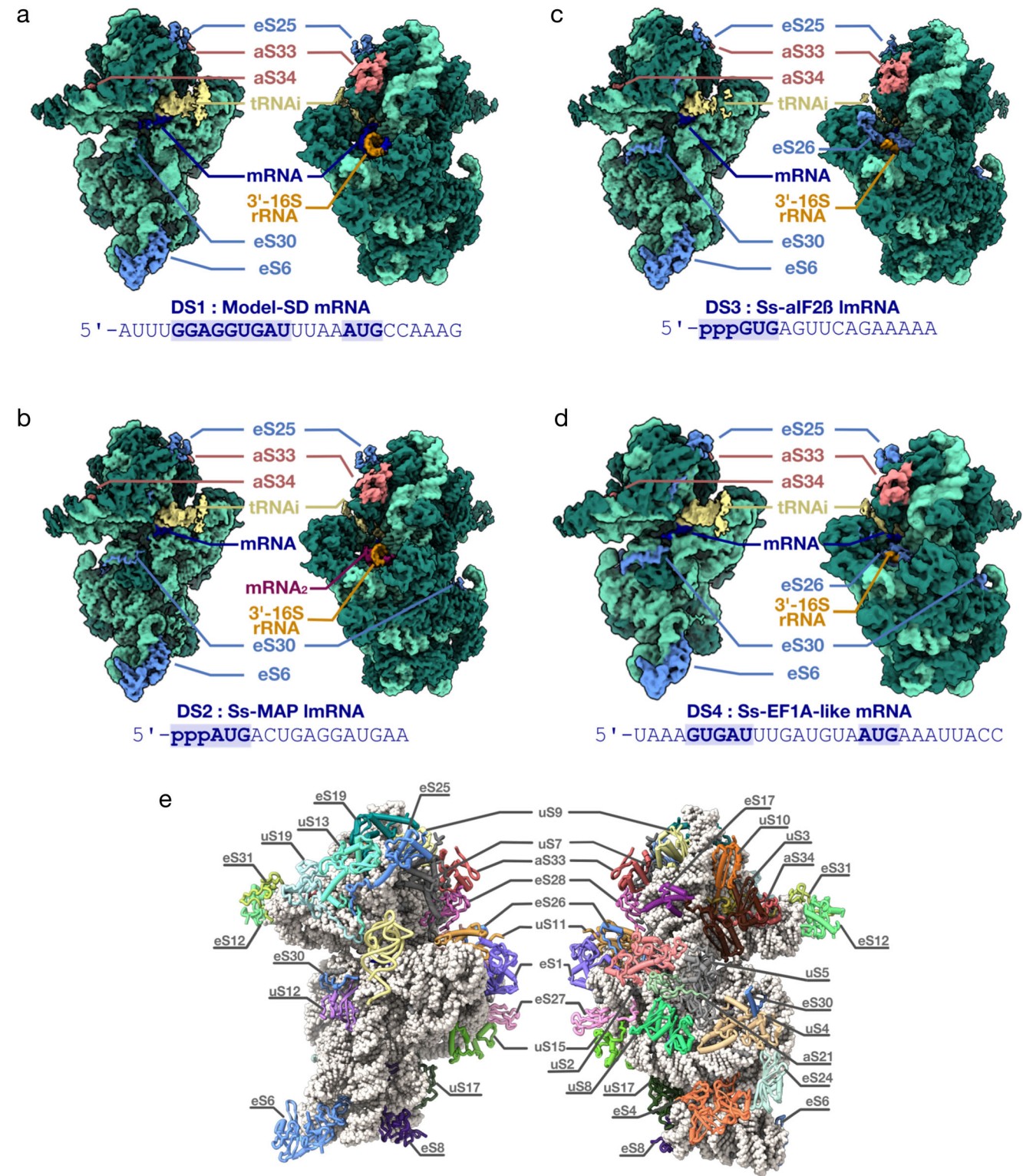

**Fig. 2 | Cryo-EM analysis of *S. solfataricus* 30S:mRNA:aIF2:Met-tRNAi^Met complexes.** **a** Cryo-EM map (3.02 Å resolution) of a 30S IC complex assembled on model-SD mRNA (DS1). The Cryo-EM map is colored according to the structural model using the volume zone command in ChimeraX[137]. The rRNA is in light green, except for the anti-SD sequence at the 3′-end which is in orange. R-proteins are in dark green except eS25, eS30 and eS6 which are in cornflower blue, and aS32, aS33 in Indian red. tRNA is in kaki and mRNA is in blue. The same color code is used in the four panels. **b** Cryo-EM map (2.8 Å resolution) of a 30S IC complex assembled on Ss-MAP lmRNA (DS2). A second mRNA molecule bound to the anti-SD sequence

(see text) is colored in magenta and labeled mRNA₂. **c** Cryo-EM map (2.94 Å resolution) of a 30S IC complex assembled on Ss-aIF2β lmRNA (DS3). **d** Cryo-EM map (3.72 Å resolution) of a 30S IC complex assembled on Ss-EF1A-like lmRNA (DS4). In views (**c**) and (**d**), eS26 bound to the mRNA exit channel is colored in cornflower blue. **e** The structure of *S. solfataricus* 30S bound to Ss-aIF2β lmRNA and Met-tRNAi^Met as observed in view (**c**). The view shows all 30S ribosomal proteins as discussed in the text. In the left view, Met-tRNAi^Met is shown in light yellow cartoon. aS33, aS34 and domain 2 of eS6 were identified in this work.

would have copurified during sample preparation but is rather due to an intrinsic mobility of helix h44. In the h44-down conformation, the classical contacts between helix h44 and the body part of the SSU are observed, as in other archaeal, bacterial and eukaryotic SSU (Supplementary fig. 6).

Unstable conformations of helix h44 were also observed in *Sulfolobus acidocaldarius* ribosomes[63] suggesting that this might be a structural feature of crenarchaeal ribosomes and explain why fully-associated ribosomal subunits are difficult to isolate in vitro[63,73,81]. Experimental conditions used here, particularly high Mg concentration (18 mM $Mg^{2+}$) and spermine (2.5 mM), could have favorably influenced the down conformation of the h44 helix by favoring contacts with the body part of the SSU.

Before discussing the characteristics of the different 30S complexes in relation to mRNA binding, we will present the specificities of the *S. solfataricus* 30S ribosomal subunit as revealed by the 2.5 Å resolution cryo-EM map calculated from 766,000 particles of DS2 (Ss-30S-HR) (Supplementary fig. 5b).

## Specific features of the small ribosomal subunit from S. solfataricus

The high-resolution structure of the *P. abyssi* 16S rRNA (PDB 7ZHG[58],) was used as a template to build the 16S rRNA of *S. solfataricus* 30S. In comparison with Pa-16S rRNA, prominent changes were observed in the sequences and structures of helices h6, h10, h16 and h44 of Ss-rRNA (Figs. 3 and Supplementary fig. 7). These regions show variability across the tree of life[19,82]. They distinguish between Thermococcales and Crenarchaeotales ribosomes (Fig. 3). More generally, they are featuring TACK and euryarchaeotal ribosomes. Interestingly, variability in these regions is accompanied by variations at the level of ribosomal proteins. For example, eS30 contacts helix h16 in *S. solfataricus* 30S whereas in Thermococcales, eS30 is not present and helix h16 is longer (Fig. 3a). On the other hand, compared with Thermococcales ribosomes, Ss-eS6 has an extra domain (12-115) and extends from the right foot to the left foot of the SSU (Fig. 3b). The additional domain (called domain 2) is inserted between the first two β strands of the N-terminal β-barrel. It corresponds to a double-ψ-β-barrel composed of two pseudo-symmetric ββαβ units (Supplementary fig. 8). This fold is considered very ancient and is conserved in many proteins[83,84]. We found homologs of Ss-eS6 proteins having domain 2 only in the Thermoprotei class, more precisely in the Acidilobales, Desulfurococcales, Fervidicoccales and Sulfolobales orders. Conversely, eS6 from archaea belonging to the other two orders of the Thermoprotei class, Thermoproteales and Thermofilales, do not possess domain 2 (Fig. 4). The presence of this domain could be linked to variations in helices h6 and h10 of rRNA. It should be noted that eukaryotic eS6 also harbors an additional domain compared with the Pa-eS6 protein. However, in eukaryotes, the additional domain is added at the C-terminus of the N-domain shared by eukaryotes and archaea (Supplementary fig. 8). This C-domain contains a very long α-helix known to be phosphorylated in most eukaryotes[85].

We identified 25 post-transcriptional rRNA modifications (Supplementary Tables 3–5 and Supplementary fig. 9). 8 of these modifications were previously mapped[86]. The number of post-transcriptional modifications identified is also in good agreement with early LC-MS studies[87]. Interestingly, the number of rRNA modifications is much less than that in the SSU of the two Thermococcales *P. abyssi* and *T. kodakarensis*[41,56,59] (Supplementary Table 3). In particular only four N4-acetylcytidines were identified. These four modified residues were confirmed using primer extension analysis (Supplementary fig. 10). Surprisingly, whereas previous studies in Thermococcales showed that N4-acetylcytidine modifications systematically targeted the second cytidine of a 5′-CCG-3′ sequence[56,59], we observed here the four N4-acetylcytidines in h45 and in two different nucleotidic contexts, $_{1465}G(ac^4C)(ac^4C)G_{1468}$ and $_{1475}(m62A)(ac^4C)(ac^4C)U_{1479}$

(Supplementary fig. 10). Moreover, among the two N(6),N(6)-dimethyladenosines expected at positions 1475 and 1476, only A1475 was evidenced in our RT experiments in agreement with the cryo-EM maps. Consistent with our observations, an aAccUG oligonucleotide (where lowercase indicates modified nucleotides), likely corresponding to the 1475-1480 segment of the 16S rRNA, was detected in ref. 86. Collectively, our data suggest the existence of specific features of the *S. solfataricus* KsgA and Nat10 modification systems. Organism-dependent modulation of the KsgA activity was already observed[88]. Moreover, evidence supporting the existence of $ac^4C$ outside of the CCG motifs were already obtained[89] suggesting that Nat10 activity could depend on adapters for substrate targeting. Finally, as all these modifications are located in close proximity, the timing of their incorporation could influence the activity of the modifying enzymes[90].

We also identified spermine residues and magnesium ions. Spermine molecules contact ribosomal proteins or rRNA or both. They contribute to 30S stabilization by creating stacking interactions with aliphatic groups and/or hydrogen bonding. They are often linked to the major groove of the rRNA, as for example in h45, where spermine interacts with modified bases (Supplementary fig. 9).

## Protein features

TACK (or Thermoproteota) and Asgard small ribosomal subunits were predicted to contain 3 additional proteins, found in all eukaryotes but not found in other archaeal phyla[21,23,32,54,56–59] (Fig. 4). In agreement with the phylogenetic studies, we observed eS25 and eS30 in all of our structures. As in eukaryotes, eS30 is located in the mRNA entry channel. The protein is well defined in cryo-EM maps of DS2-4 with only the 8 last residues of the protein not visible. Interestingly, the N-terminal extremity of eS30 contacts the h44 helix. Van-der-Waals interactions occur between Pro2 (expected to be the first amino acid of archaeal version of eS30 after methionine excision) and the glycosidic bond linking G1448 to A1449 (Fig. 5a). Interestingly, most archaeal eS30 have a Pro in position 2, whereas a Pro at this position is never observed in eukaryotes (Supplementary fig. 11). In eukaryotic translational decoding complexes, the N-terminal extremity of eS30 participates in the stabilization of the codon:anticodon interaction at the A site, via a conserved histidine residue (Supplementary fig. 11 and Supplementary Data 1)[66,91] and also interacts with eEF2 during translocation[92]. In addition, in eukaryotic initiation complexes, eIF1A is bound to the A site and contacts the N-terminal tail of eS30, including residues 2 to 8[37]. Therefore, studies in eukaryotes show a role of the N-terminal tail of eS30 in translation processes. Here, the positioning of the Ss-eS30 N-terminal tail close to helix h44 is not changed whatever the mRNA or the concentration of aIF1A used in our complex reconstitutions, and aIF1A is never observed. It is possible that a local conformational change at the 30S A site is required to trigger displacement of the Ss-eS30 N-terminal tail and binding of aIF1A.

Ss-eS25 is composed of a core domain and an elongated N-terminal part (Figs. 5a and Supplementary fig. 12 and Supplementary Data 1). The core domain is located at the top of the 30S head, as in eukaryotic SSUs. Region 12-25 was modeled according to the Alpha-Fold2 prediction[93] and tentatively positioned in the cryo-EM maps. The N-terminal peptide, from Gly2 to Thr11, is well defined in the cryo-EM map. Interestingly, this peptide interacts with region 132-138 of uS13 located close to the P site initiator tRNA (Fig. 5a). Sequence alignments of archaeal eS25 reveal strong conservation of the two N-terminal glycine residues involved in the interaction with uS13. In contrast, more variability is observed in eukaryotic eS25 that never have two glycines (Supplementary fig. 12). Notably, a direct interaction of the N-tail of eS25 and the P site tRNA was observed in human ribosomal elongation complexes[66].

Ss-eS26 is not visible in all our 30S translation initiation complexes but, as discussed below, its presence depends on the occupancy of the mRNA exit channel.

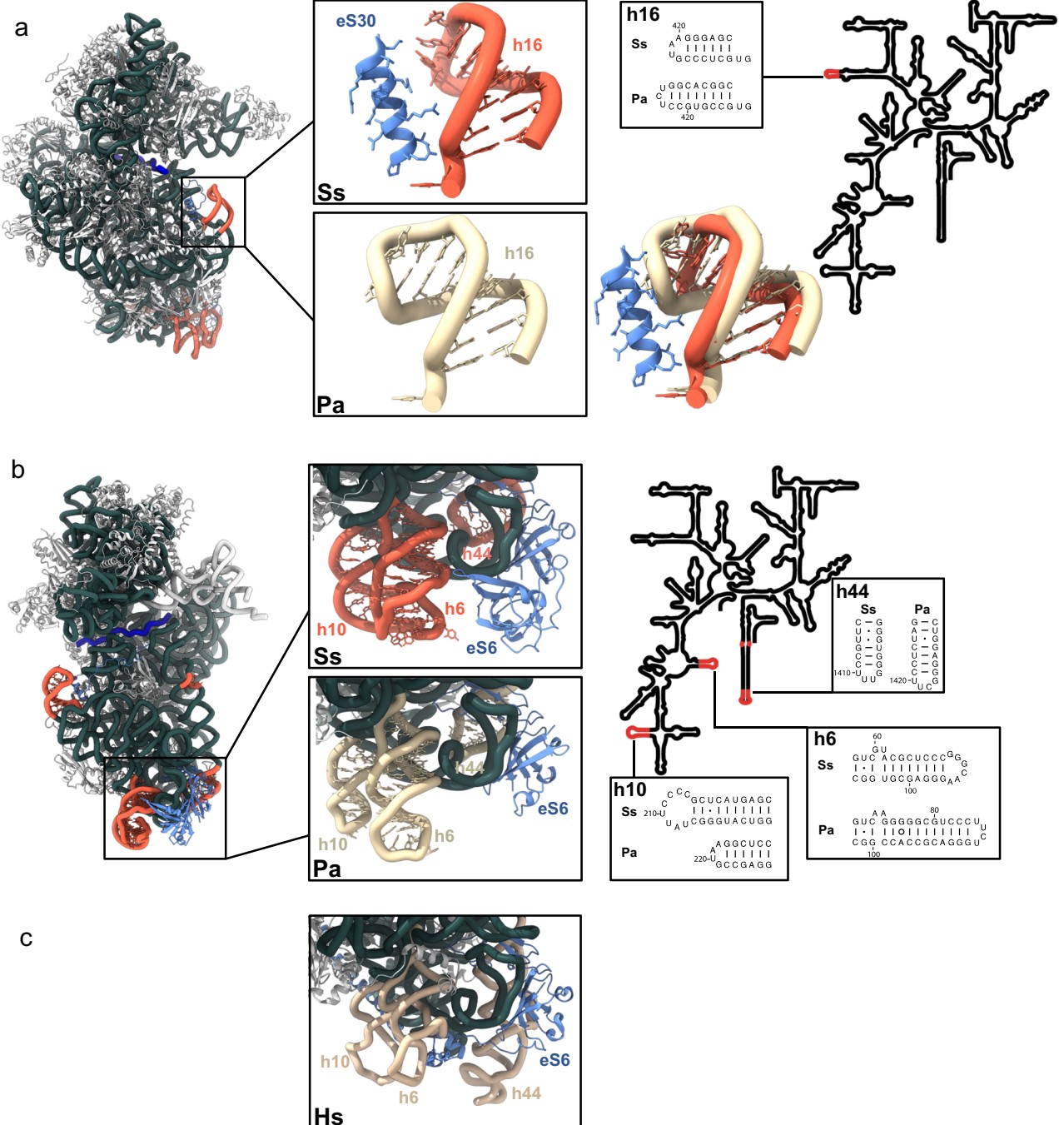

**Fig. 3 | Comparison of 16S rRNAs from *S. solfataricus* and *P. abyssi*. a** h16 and eS30 binding site. The cryo-EM structure of Ss-30S is shown on the left with the rRNAs helices of interest colored in red (*S. solfataricus*) or in light yellow (*P. abyssi*). eS6 is colored in cornflower blue. **b** Region of h10, h6, h44 and eS6 (cornflower blue) binding site. **c** Position of eS6 in the human ribosome (8QOI[70]). See also Supplementary fig. 8 for a description of eS6 in eukaryotes and archaea.

Besides eS25, eS26 and eS30, the cryo-EM structure also revealed the presence of two supplementary proteins that were not annotated as ribosomal proteins[33]. One of these proteins is located in the beak of the SSU (Fig. 5b). A first tracing of its backbone in the cryo-EM map allowed us to identify a *ca* 80 residue protein containing 2 zinc binding motifs, 7 cysteines and one tryptophane. Then, a search in UniProt for uncharacterized *S. solfataricus* proteins identified the ORF. The second unexpected protein was found behind the SSU head. The corresponding ORF in the *S. solfataricus* genome was identified using the strategy described above. These two ribosomal proteins are specific to archaea Thermoprotei. We could not find convincing bacterial or eukaryotic homologs. Because of their "exclusively archaeal"

character, we propose to name these two proteins aS33 (UniProt D0KTI0) and aS34 (UniProt Q97ZK4) (Fig. 4). The presence of the two proteins in our 30S preparation was confirmed by mass spectrometry (Supplementary fig. 2). Interestingly, the same phylogenetic distribution was observed with aS34 and domain 2 of eS6. The same results were also obtained for aS33, except that a few homologs were also found in Candidatus Marsarchaeota, Bathyarchaeota and Culexarchaeota. Thus, domain 2 of eS6, aS34 and aS33 strongly co-occur (Fig. 4 and Supplementary Data 1). In the *S. solfataricus* genome the three genes are not organized in a single operon.

aS33 binding site is located close to the mRNA exit channel (Figs. 5, 6, Supplementary fig. 13). It interacts with the minor groove of

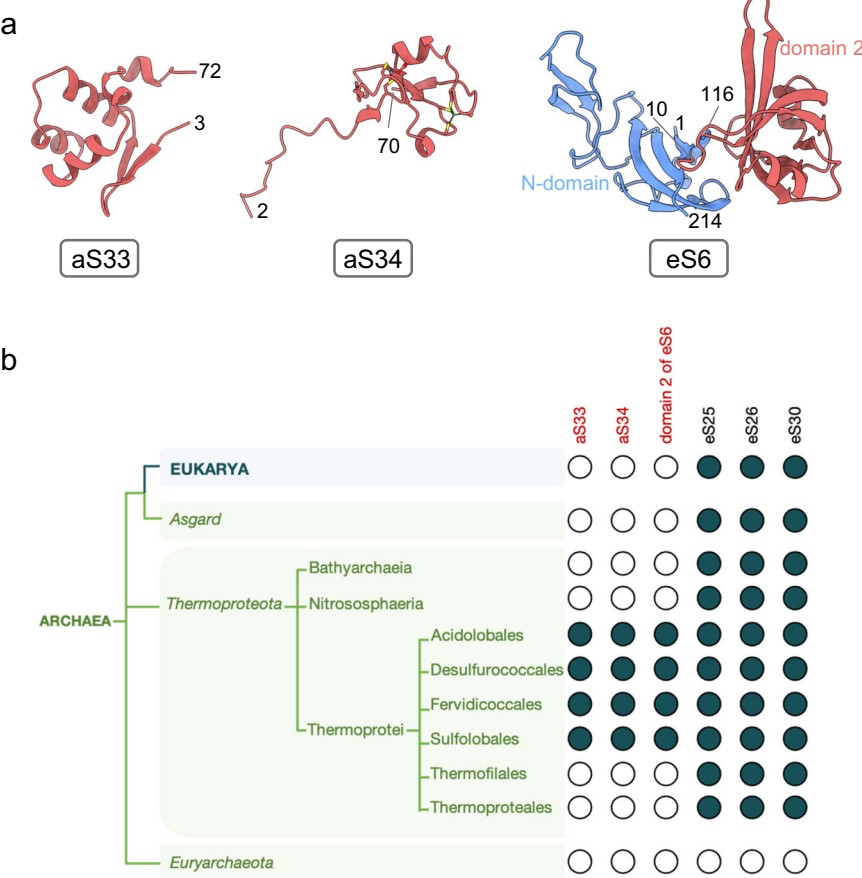

**Fig. 4 | 3D structures and phylogenetic distribution of Ss-aS33, Ss-aS34 and Ss-eS6-domain 2. a** 3D structures of aS33, aS34 and eS6 as observed here in *S. sol-fataricus* 30S. eS6 is composed of two domains (see also Supplementary fig. 8). The N-domain is present in all eukaryotes and archaea, while domain 2 is present only in Thermoprotei (see below). **b** BlastP and Hmmer searches show that of Ss-aS33, Ss-aS34, Ss-eS6-domain 2 orthologs are found only in Thermoprotei and more precisely in the Acidilobales, Desulfurococcales, Fervidicoccales and Sulfolobales orders. Ss-aS33, Ss-aS34, Ss-eS6-domain 2 are not found in Thermoproteales and Thermofilales. Note that eS6 N-domain is present in all archaea.

h40 and with uS7, eS28 and uS9. Its binding site overlaps that of eIF3 subunit d as observed in mammalian 48S late-stage translation initiation complex[39] and in *Trypanosoma cruzi*[38]. Notably, eIF3d is involved in noncanonical cap-dependent translation of specific mRNAs[94,95]. Search for structural similarities using Foldseek[96] revealed significant homology with one ORF of *Sulfolobus islandicus* rudivirus 1[97] suggesting that viruses may have contributed to disseminate this protein in Thermoprotei. The fold is also found in prediction of human UFL1, a protein involved in UFMylation and in DNA binding.

aS34 interacts with h33, uS3 and uS14 (Fig. 5b). It occupies a position equivalent to that of eS10 in the eukaryotic ribosome but no structural homology is observed. Interestingly, eS10 and uS10 located in close proximity, are the targets of ubiquitination during mammalian ribosome quality control mechanisms[98,99]. Search for structural similarities using Foldseek[96] revealed some similarity of aS34 with ring E3 ubiquitin protein ligase domain that also bears two zinc ions[100] (Supplementary fig. 14). A possible role for this protein in ribosome quality control remains to be investigated.

### Initiation complex with an SD-containing mRNA
One initiation complex was prepared using an mRNA containing a strong SD sequence (Fig. 2a). The IC was prepared by mixing 30S, mRNA, aIF1A and aIF2:tRNA and then purified by gel filtration on a Bio SEC-5 HPLC column[101] (Methods and Supplementary fig. 4b). The complex was then used for data collection on a Titan Krios microscope (DS1, Supplementary fig. 5a). As described above, 3D classification

enabled us to sort out h44-up and -down conformations. Two intermediate biogenesis states were also isolated. Their analysis is beyond the scope of the present article and they will therefore be described elsewhere. Nevertheless, the isolation of these two states shows that a fraction of immature ribosomes is present in our ribosome preparation.

IC h44-down structure shows the initiator tRNA in the P site, base-paired with the AUG start codon (Fig. 7a). aIF2 is bound to the 3'-methylonylated end of the tRNA. However, the cryo-EM map for aIF2 is fuzzy suggesting that the factor is highly mobile in the solvent side and does not tightly interact with the ribosome. The codon:anticodon duplex is stabilized by interactions with the C-terminal tails of uS9, uS13 and uS19 as observed in Thermococcales[56,58]. Whereas the C-terminal tails of uS9 and uS19 are conserved in all archaea, the C-terminal end of uS13 has an extension (17 aa in *S. solfataricus*) specific to Crenarchaeota[21]. However, this extension is not visible in the cryo-EM map. A set of modified rRNA bases participates in the stabilization of the codon:anticodon interaction (Fig. 7a). In particular, C34 is stacked onto hypermodified N1-methyl-N3-(3-amino-3-carboxypropyl) pseudouridine 930 located at the tip of helix 31. Notably, the carboxyl group of $m^1acp^3\Psi$ interacts with the amino group of C1364 located opposite. In *P. abyssi*, the N3-(3-amino-3-carboxypropyl) modification is not found and a $m^5C$ is found in place of *S. solfataricus* C1364. The presence of a methyl group at this position would create a conflict with the $acp^3$ chemical group. This shows that the set of modifications evolved differently in the Thermococcales and in the Thermoprotei. In

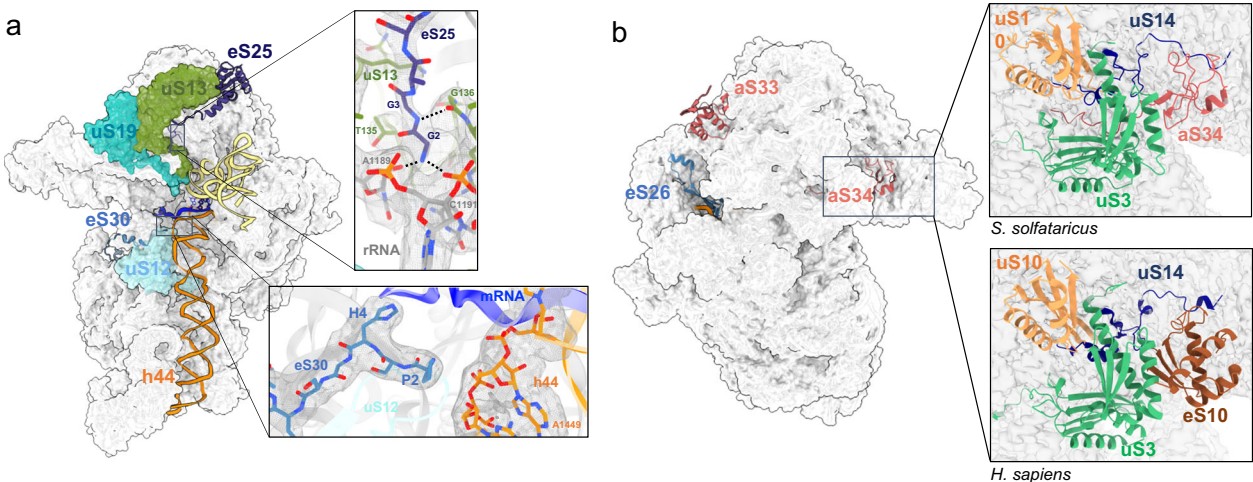

**Fig. 5 | Archaeal versions of eS25, eS26, eS30 and herein identified 30S ribosomal proteins. a** Cartoon representation showing the positions of eS25 and eS30. Two close-up views show the DS2 cryo-EM map around the N-tails of eS30 and eS25. The cryo-EM map suggests interaction of eS25-Gly2 with residues A1189 and C1191 and the carbonyl group of uS13-T135 as well as an interaction between eS25-G3 and the carbonyl group of uS13-G136. **b** Same as part A but the view is rotated by 180°, showing the positions of eS26, aS33 and aS34. On the right, the close-up views compare the *S. solfataricus* case (top) with the human case (bottom; PDB 8G5Y[66] or 8QOI[70]). Note that eS26 is only visible in IC-DS3 and IC-DS4.

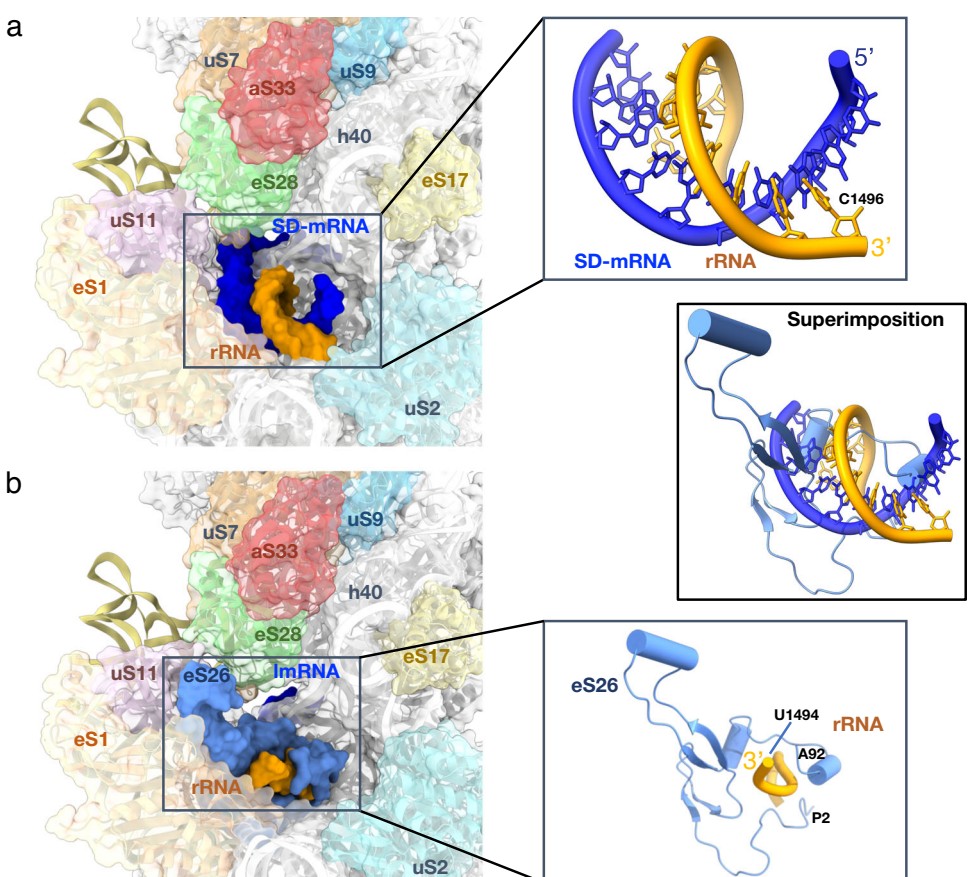

**Fig. 6 | mRNA exit channel. a** View of the mRNA exit channel of IC-DS1. The SD duplex is bound to the channel. The mRNA is in dark blue and the rRNA is in orange. Peripheral proteins are indicated. **b** View of the mRNA exit channel of IC-DS3. eS26 (cornflower blue) is bound to the channel wrapped around the 3' end of the rRNA. Superimposition of the two structures shows that SD duplex and eS26 bindings are not compatible.

agreement with the identification of m¹acp³Ψ modification, genes coding for Tsr3 and Nep1, known to be responsible for its biogenesis, are found in *S. solfataricus*[102–105]. The presence of the modification at this position also likely reflects the correct completion of the final stages of ribosome biogenesis[105]. Notably, m¹acp³Ψ is also found at the same position in eukaryotes and shown to be important for the translocation mechanism[92].

The path of model-SD mRNA towards the exit channel corresponds to that observed in *P. abyssi* IC[41,56]. In the mRNA exit chamber, the Shine-Dalgarno sequence of the mRNA is base-paired with the

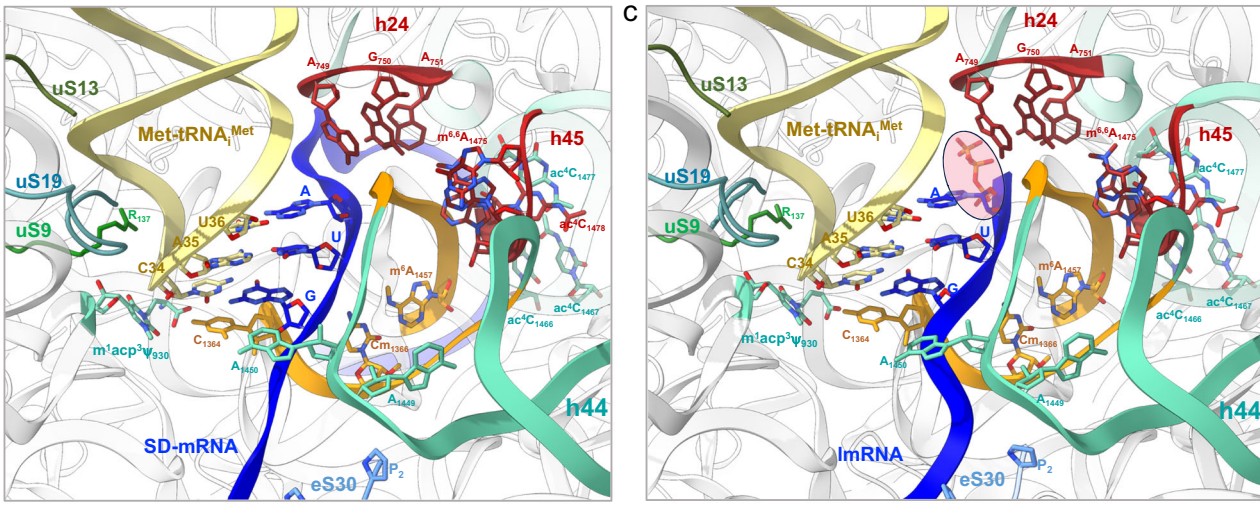

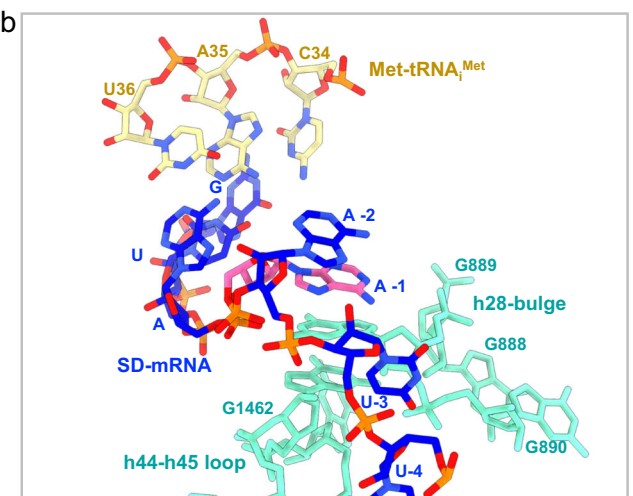

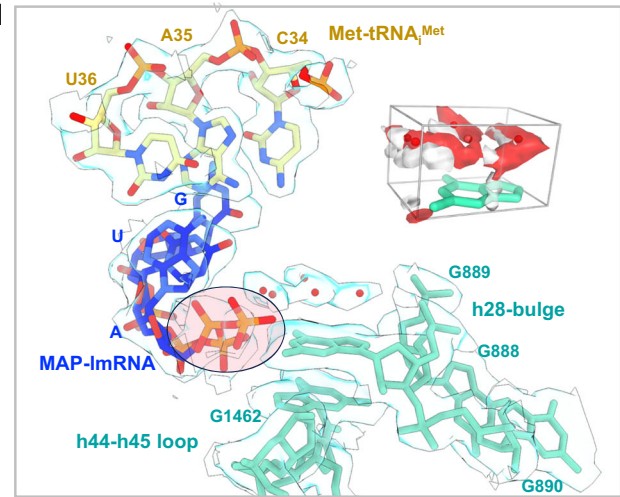

**Fig. 7 | Interactions at the P site. a** Network of interactions at the P site in IC-DS1. The proximal part of the h44 helix is in orange. The h24 loop and parts of h45 interacting with h44 are in red. mRNA is in blue and the initiator tRNA is in yellow. Modified nucleotides are colored by atoms and labeled. Parts of the rRNA located close to the codon-anticodon duplex are in aquamarine. The same color code is used in the three views. **b** Close-up of the codon:anticodon interaction in IC-DS1. **c** Same as view a but for IC-DS2. **d** Same as view b but for IC-DS2. The cryo-EM map is also shown to highlight the network of water molecules located above G889. The view shows that the water network mimics base −1, as observed in the leadered mRNA shown in view b (base −1 in pink). A zoomed view on the region above G889 nucleobase studied by molecular dynamics is shown in the right-hand corner. The view shows that the GIST predicted water oxygens (red) and hydrogen (white) densities fit with the experimentally placed water molecules (red spheres) (see also "Methods" and Supplementary figs. 15, 16).

3′ end of the 16S rRNA (Fig. 6a). Interestingly, the cryo-EM map suggested a 5′-CCUCC-3′ sequence at the 3′ end of the 16S rRNA whereas the DNA sequence (*S. solfataricus* P2 strain, GenBank AE006641) is 5′-CCTCA-3′. To avoid any misinterpretation of the cryo-EM map, we determined the 16S DNA sequence of the P2 strain used for our ribosome preparations (Fig. 8a) as well as that of the 16S rRNA (Fig. 8c). The 5′ end of the 16S rRNA was first mapped by reverse transcription (Fig. 8b). Next, we circularized 16S rRNA and sequenced the region corresponding to the 5′-3′ junction to identify the 3′ end of the rRNA. This analysis confirmed that the 3′end of the 16S rRNA is 5′-CCUCC-3′ as anticipated from the cryo-EM map, whereas the gene sequence was 5′-CCTCA-3′, as expected from the published genomic sequence. This result suggests that the 3′ end of the 16S rRNA is modified during ribosome biogenesis. In line with this idea, we noticed faint additional bands in the 16S rRNA sequencing reaction, suggesting possible heterogeneity. We therefore amplified and cloned the region corresponding to the 5′ and 3′ junction. The sequences of 32 individual clones were determined. As shown in Fig. 8, 27 clones showed a 5′-CCUCC sequence, 1 showed the 5′-CCUCA sequence and 4 clones

showed an intermediate situation, 3 with a 5′-CCUC sequence and 1 with a 5′-CCU sequence. Overall, this analysis strongly argues in favor of a processing of the 3′ extremity of the rRNA that converts the originally transcribed 5′-CCTCA into 5′-CCUCC.

Transcriptomic data[48] showed that 73 % of mRNAs are leaderless (5′UTR ≤ 5 nucleotides), 11 % of the mRNAs have a 5′UTR with 6 to 10 nucleotides and 16 % have a leader regions with more than 11 nucleotides (leadered mRNAs). Moreover, 879 genes were identified as distal cistrons. We compared the exact occurrences of GGAGG and UGAGG sequences upstream from start codons. For the 169 leadered proximal cistrons, we found 9 GGAGG (5.3%) and 7 UGAGG (4.1%). The difference is more pronounced for the 879 distal cistrons with 72 GGAGG (8.2%) and 47 UGAGG (5.3%). Although the impact of the A to C modification at the 5′ end of the 16S rRNA is not clear at this stage, a possible role in translation reinitiation at distal cistrons can be envisaged.

## Binding of leaderless mRNAs
We then wanted to obtain information on leaderless mRNA binding. We used Ss-MAP lmRNA and Ss-aIF2β lmRNA (Supplementary Table 1).

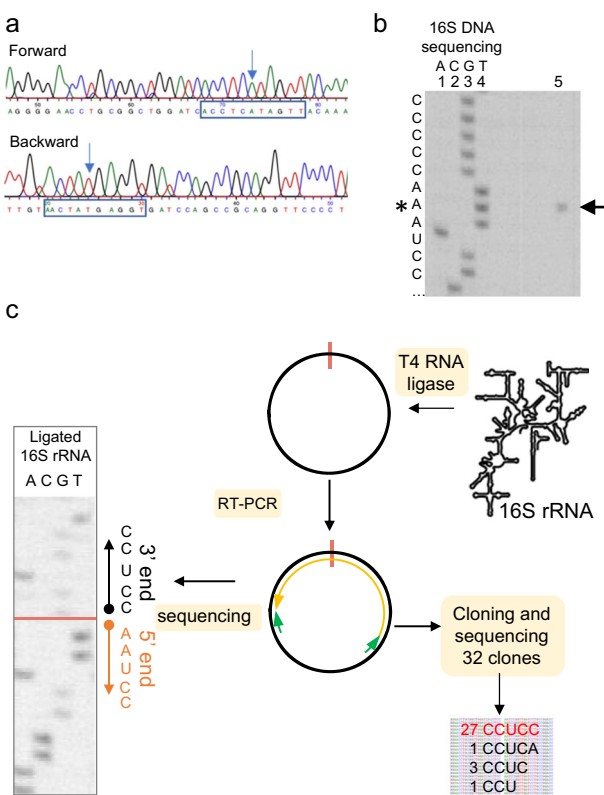

**Fig. 8 | Determination of the 3′ and 5′ ends of the 16S rRNA. a** DNA sequencing electrophoregrams of the amplified 16S gene from *S. solfataricus* cells. Regions around the putative 3′-end of the 16S rRNA are boxed. 16S rDNA shows a CCTCA sequence. **b** Mapping of the 5′-end of the 16S rRNA using reverse transcription (RT). Lanes 1 to 4: DNA sequencing ladder of the amplified 16S gene using the reverse transcription primer. The complementary RNA sequence is shown on the left. Line 5: reverse transcription analysis of the 16S rRNA using the 5′ mapping reverse transcription primer. The arrest is indicated by an arrow. This experiment shows that the 5′-end of the 16S rRNA starts with the 5′-AAUCC sequence. The first base is indicated by a star. This experiment was performed twice (*n* = 2). **c** Identification of the 3′-end of the 16S rRNA. The 16S rRNA was first circularized using T4 RNA ligase, RT-PCR amplified (*n* = 1) and sequenced (*n* = 2). Left: sequencing of the RT-PCR amplified ligated 16S rRNA fragment. This qualitative experiment identified that the major sequence of the 3′ end is CCUCC. This result is consistent with the rRNA oligonucleotide catalogs that were early published[86]. In parallel, a XhoI·BamHI fragment from the RT-PCR material, containing the 3′ and 5′ regions, was cloned into a pBS plasmid. Thirty-two independent clones were sequenced. The results show heterogeneity in the length of the 16S rRNA, as discussed in the text. Source data are provided as a Source Data file.

The initiation complexes, assembled in the presence of a 10-fold (relative to 30S) excess of eS26, were affinity purified using His-tags at the N-termini of aIF2β and aIF2α subunits. Datasets 2 and 3 were collected on Titan Krios microscopes (Supplementary fig. 5). Following the same strategy as described above, we identified 17% and 31 % of the particles corresponding to an IC h44-down conformation in DS2 and DS3. In both DS2 and DS3, the leaderless mRNA is bound to the P site and the start codon is base-paired with the Met-tRNA$_i^{Met}$ anticodon. A network of interactions involving uS13, uS19 and uS9 participates in codon:anticodon stabilization in a manner similar to what was observed with the SD-leadered mRNA (Fig. 7c). The three phosphate groups are observed at the 5′ end of the lmRNAs. Strong density exists for the α and β phosphates showing that they are tightly bound. The density for the gamma phosphate is less unambiguous suggesting two alternative conformations in DS2. Interestingly, water molecules linked to α and β phosphates create water-nucleobase stacking contacts with G889 (Fig. 7d and Supplementary fig. 15a). Overall, the triphosphate

group and the network of water molecules mimic the position of a -1 base as observed in DS1, contributing to the binding of the lmRNA (Fig. 7c, d).

We used explicit water molecular dynamics simulation, followed by grid inhomogeneous solvation theory (GIST) analysis to confirm the propensity of water molecules to interact with G889[106,107] (see Supplementary Methods). GIST is a method giving access to the structure and thermodynamics of solvent in the vicinity of a solute molecule. As described in Supplementary information, our results support the assignment of the five water molecules in the region above G889 (Figs. 7d, Supplementary fig. 16 and Supplementary Data 1). Favorable interaction energies of these water molecules with G889 are calculated. These water positions are compatible with OH and/or lone pair stacking interactions with G889, as studied by quantum chemical calculations[108]. At last, we note that the shape of water density above G889 nucleobase is suggestive of a purine nucleobase, where the cycle atoms would be replaced by a network of water molecules.

The importance of the triphosphate group in lmRNA binding was also studied in toeprinting experiments with native tri-phosphorylated Ss-aIF2β lmRNA, de-phosphorylated Ss-aIF2β lmRNA and re-phosphorylated (monophosphate) Ss-aIF2β lmRNA (Methods). As shown in Fig. 1b, the toeprint signal decreased when Ss-aIF2β lmRNA was de-phosphorylated. Moreover, we observed an intermediate effect with monophosphorylated Ss-aIF2β lmRNA. In contrast, no influence of the 5′ triphosphate group on the toeprinting signal was observed with the model-SD leadered mRNA (Supplementary fig. 17). Overall, our results show that the 5′ triphosphate group participates in the stabilization of a 30S IC formed with a leaderless mRNA.

Finally, Ss-aIF2β lmRNA starts with a GUG codon whereas Ss-MAP lmRNA starts with an AUG codon. Superimposition of the two structures shows a slight difference in the orientation interaction of the first base pair, but interpretation is limited by the resolution of the cryo-EM maps (Supplementary fig. 15b–d). This adjustment may illustrate how the nature of the start codon can influence the binding affinity of the lmRNA and its translation efficiency.

In Ss-MAP lmRNA IC structure, the E-site is empty, as expected. However, the mRNA exit chamber is occupied by a duplex involving the 3′ end of the 16S rRNA and an RNA. We realized that the sequence of the Ss-MAP lmRNA contained a pseudo-SD sequence downstream from the start codon (Supplementary Table 1) allowing base-pairing with the anti-SD sequence (Fig. 2b). Thus, two Ss-MAP lmRNA molecules are bound simultaneously, one at the P site and an additional one forming a pseudo-SD duplex (Supplementary fig. 18). In DS2, we also identified a fraction of particles where no binding of P site tRNA and mRNA was observed, neither at the P site nor in the exit channel (Supplementary fig. 5). In these particles, eS26 is visible in the mRNA exit channel.

In Ss-aIF2β lmRNA IC, the E site is empty and eS26 is bound to the exit channel. eS26 is wrapped around the 3′ single stranded extremity of the 16S rRNA (G1486 to C1493). eS26 also interacts with uS11 and eS1 and contacts rRNA of the head and of the platform domains (Fig. 6b). Superimposition of the IC-DS3 structure onto IC-DS1 or IC-DS2 shows that the binding of SD:antiSD duplex to the mRNA exit chamber is not compatible with eS26 binding (Fig. 6). Consequently, either eS26 is released from the SSU upon binding of mRNA containing SD sequences or the ribosomal fraction bound to eS26 does not bind SD-mRNAs. Remarkably, the position of eS26 observed in the archaeal SSU exactly corresponds to the position of eS26 observed in yeast and human ribosomal complexes[38,39,66,70] (Supplementary fig. 19). Competition between eS26 and mRNA binding in the exit channel was also illustrated in toeprinting experiments. We observed that high concentrations of eS26 led to disappearance of the main toeprinting signal with leadered mRNAs, whereas they had no impact on the toeprinting signal observed with a leaderless mRNA (Fig. 1c and Supplementary fig. 20). Moreover, with the leadered Ss-EF1A-like mRNA, high

concentrations of eS26 also increased an alternative toeprinting signal corresponding to a GUG codon located at seven nucleotides from the 5′ end. Therefore, eS26 may favor leaderless mRNA translation initiation. To test this hypothesis, we determined the cryo-EM structure of an initiation complex bound to a 28-nucleotide Ss-EF1A-like mRNA prepared in the presence of a 10-fold excess of eS26 (DS4, Supplementary fig. 5). The cryo-EM map shows an mRNA base paired to the initiator tRNA. The E site codon and perhaps one additional mRNA base are visible, but no more. In addition, eS26 is bound to the exit channel at the 3′ end of the 16S rRNA. In agreement with our toeprinting experiment, it appears that eS26 has prevented mRNA binding to the exit channel. Consequently, the mRNA was bound to the 30S as an mRNA carrying a short 4-nucleotide leader with a GUG codon base-paired to the initiator tRNA.

At this stage, two possible scenarios exist to explain eS26 function. A first possibility is that there exists a fraction of ribosomes that contains eS26 and that this specialized fraction is responsible for lmRNA translation. Alternatively, binding of eS26 to the ribosome might be dynamic and compete with SD-mRNA binding. Our mass spectrometry analysis suggests that eS26 is sub-stoichiometrically present in our Ss-30S preparations (Supplementary fig. 2). Consistent with this observation, western blotting experiments determined that only a small fraction of our 30S contains eS26 (~5%, See Supplementary fig. 21a). In the same view, cryo-EM data collections using vacant 30S ribosomes followed by focused classification did not allow us to identify a class where eS26 is present (Supplementary fig. 21b). These results indicate that only a small fraction of 30S in our preparations contains eS26. Finally, we show using a toeprinting experiment (Supplementary fig. 22) that the model-SD mRNA is able to displace eS26 when it was previously bound to Ss-30S. Taken together, our data strongly suggest that eS26 binding to Ss-30S is reversible rather than biochemically stable.

## Discussion

At a time when the archaeal diversity is being revealed and phylogenetic studies are discussing the possible origin of eukaryotes within an archaeal branch, functional studies of core information processing machineries of archaea are strongly needed. Similarities between the molecular mechanisms in archaea and eukaryotes are apparent in studies conducted with *Saccharolobus*/*Sulfolobus* cells, which are currently the most experimentally accessible members of the TACK, sister group of the Asgardarchaeota and Eukarya (see[109] for review). Here, we unveil features of the translational machinery of the archaeon *S. solfataricus*, belonging to the TACK superphylum. Among them, two ribosomal proteins, aS33 and aS34, not previously annotated, were evidenced. These two proteins, as well as an additional domain of eS6, are specific to Thermoprotei. Comparisons of the Ss-30S with eukaryotic SSUs as well as structural features of aS33 and aS34 suggest that the recruitment of these proteins to the ribosome might be related to the acquisition of additional functions possibly in mRNA binding (aS33) or ribosome quality control (aS34). In addition, our study reveals an unexpected maturation at the 3′ end of 16S rRNA that results in a CCUCC 3′end rRNA sequence while the corresponding DNA sequence is CCTCA. In agreement with our findings, a 3′ terminal CCUCC sequence was mentioned in oligonucleotide catalogs derived from 16S rRNA of *S. solfataricus* and *S. acidocaldarius*[86]. However, as the DNA sequences of the genes were not known at the time, possible processing of the 3′ end was not suspected.

Our study also provides information about the archaeal versions of the three ribosomal proteins eS25, eS30 and eS26, only found in TACK and Asgard (Fig. 4). These proteins are known to be directly involved in translation mechanisms in eukaryotes. Here, eS25 and eS30 are observed in all of our complexes. The core domains of these ribosomal proteins are located at the same sites as their eukaryotic counterparts. In addition, Ss-eS25 interacts via its N-terminus with uS13

that itself interacts with the P site tRNA. The N-terminus of Ss-eS30 is seen near to the A site, interacting tightly with helix h44 (Fig. 4). These observations strongly suggest a direct role of Ss-eS25 and Ss-eS30 in archaeal translation. In human, eS25 directly interacts with the P site tRNA during translation via its N-terminal tail and the N-terminal tail of eS30 contacts the A site tRNA during decoding[66,91]. Interestingly, sequences of eS25 and eS30 N-tails harbor eukaryotic or archaeal signatures likely reflecting specificities in translation mechanisms (Supplementary figs. 11 and 12).

The case of Ss-eS26 is particularly intriguing. When Ss-eS26 is present, it is bound to the mRNA exit channel and wrapped around the 3′ end of the 16S rRNA, interacting with it as this is observed in eukaryotes (Figs. 6 and Supplementary fig. 19). Binding of eS26 to the mRNA exit channel is not compatible with that of an SD:antiSD duplex (Figs. 2 and 6). Furthermore, using Ss-EF1A-like mRNA, we observed competition between eS26 and the mRNA for binding to the exit channel. Excess of eS26 prevents accommodation of the SD sequence of the mRNA in the exit channel and it becomes bound as an lmRNA (Supplementary fig. 20). Thus, by competing with leadered mRNA, eS26 favors leaderless mRNA translation.

Our data indicate that eS26 is present in only a small fraction of our Ss-30S (Supplementary fig. 21). As 30S subunits were purified prior to analysis, it is not possible to firmly conclude about the fraction of 30S subunit containing eS26 in vivo. However, our data show that eS26 does not bind 30S as tightly as other ribosomal proteins. This is evidenced by the low occupancy of the eS26 site in purified Ss-30S observed by both Western blot and cryo-EM analysis of vacant 30S. Moreover, the eS26 site can be readily populated by adding an excess of the protein, as shown in toeprinting experiments (Supplementary fig. 20 and 22) and in DS3-4 cryo-EM structures. The whole data render unlikely the hypothesis that specialized ribosomes stably containing eS26 specifically initiate translation of lmRNAs. Rather, our data support the idea that eS26 acts as an initiation factor favoring translation initiation at lmRNAs at least by impairing SD-leadered mRNAs binding. Consistent with this idea, the transcriptomic study[48] shows a high number of reads supporting eS26 mRNA transcription start, suggesting that the protein is abundant. Thus, eS26 concentration might regulate lmRNA translation initiation in *S. solfataricus*. This role is particularly important in an organism where lmRNAs and SD-leadered mRNAs coexist, with SD-leadered mRNAs efficiently binding the anti-SD sequence and lmRNAs representing a major fraction of mRNAs.

In eukaryotes, eS26 is a bona fide ribosomal protein whose incorporation is directly linked to the final stages of biogenesis and to the action of its chaperone Tsr2[78,79]. eS26 was shown to contribute to the translation of specific mRNAs through recognition of the Kozak sequence[110,111]. Furthermore, stress conditions due to high salt or high pH induce specialized ribosomes lacking eS26, which lose mRNA sequence selectivity and promote stress response. The Tsr2 protein, by binding to eS26, would facilitate its release from ribosomes to enable this reversible response[111,112]. Eukaryotic eS26 has a C-terminal extension of around 15 residues[113] (Supplementary fig. 19). This extension is important for its binding to mRNAs, as shown by cryo-EM structures[37,39]. Interestingly, an important increase in translation of mRNAs with short 5′UTR bearing TISU elements (Translation Initiator of Short 5′UTR) was observed in cells expressing an eS26 version lacking its C-terminal tail and thus even more closely resembling the archaeal protein[114]. Taken together, these studies point to a regulatory role for eS26 in mRNA binding during eukaryotic translation, reminiscent of our observations. The eukaryotic specificities of eS26 could be linked to the acquisition of the C-terminal tail during evolution.

In archaea, eS26 could also influence translation initiation at the distal cistrons. In this context, we noted that when present, the Shine-Dalgarno sequences on leadered mRNAs have more GGAGG sequences in the distal cistrons than in the proximal ones. The modification of the sequence at the 3′-end of the rRNA could contribute to the SD:antiSD

interaction and facilitate the assembly of the initiation complex at the start codon of distal cistrons. Finally, our results on Ss-eS26 are also reminiscent of those obtained with bacteroidetes where the bS21 protein, along with bS18 and bS6, sequesters the 3' end of 16S rRNA[69,115]. Bacteroidetes prefer to use mRNAs devoid of SD sequences and may use Shine-Dalgarno sequences as regulatory elements. Sequestration of the 3' of the 16S rRNA would then be crucial to the recruitment of SD-free mRNAs.

Our studies show that 30S:mRNA complexes can be assembled using leaderless mRNAs and that their assembly is strongly stabilized by the initiator tRNA and aIF2. We show that the 5'tri-phosphate group of lmRNAs helps stabilize the codon:anticodon interaction at the P site by mimicking the base at position -1 of leadered mRNAs thanks to a network of water molecules (Fig. 7d). The importance of the triphosphate group in the stability of the initiation complex is further validated by toeprinting experiments showing that a 3'-hydroxyl group disfavors IC stability (Fig. 1). This result is reminiscent of the bacterial case where the importance of the 5' triphosphate group at lmRNAs was previously shown[116]. However, because bacterial ribosome and initiation factors differ from the archaeal ones, lmRNA translation initiation mechanisms are expected to be distinct[117,118].

On the other hand, some Euryarchaeota, for instance *Haloferax volcanii*, also mainly use lmRNAs[51,119]. These archaea do not have eS26. Moreover, although a fraction of the genes contain SD sequences, their functionality was questioned[120]. It is therefore likely that lmRNA translation initiation has evolved differently in Euryarchaeota as compared to the TACK and Asgard branches.

To conclude, our study provides illustration of archaeal versions of eS25, eS26 and eS30 present in Asgard and TACK archaea, the closest relatives of eukaryotes. The binding of eS26 to the mRNA exit channel, wrapped around the 3' end of the rRNA, as it is observed in eukaryotes, is incompatible with SD duplex binding. The position of eS26 in the mRNA exit channel and that of eS30 close to the mRNA entry channel may reflect evolution of the mRNA recruitment leading to the mechanisms observed in eukaryotes.

## Methods

### Growth of *S. solfataricus* and preparation of ribosomal subunits

*S. solfataricus* cells were grown at 78 °C in 1 L flasks essentially as described[121]. Ribosomal particles were purified using experimental conditions described in refs. 72–74 with some modifications. Three grams of cell pellets were resuspended in 16 mL of buffer A (20 mM MOPS pH 6.7, 10 mM $NH_4Cl$, 18 mM magnesium acetate, 0.1 mM EDTA, 6 mM 2-mercaptoethanol, 2.5 mM spermine) containing 0.1 mM PMSF and 0.1 mM benzamidine. 16 mL of glass beads (5 μm, Sigma) were added and cells were disrupted by using a Vibrogen-Zellmühle shaker (Edmund Bühler GmbH, Germany). After extensive washing of the glass beads and centrifugations, the crude extract (34 mL) was loaded onto a 35 mL sucrose cushion (20 mM MOPS pH 6.7, 10 mM $NH_4Cl$, 18 mM magnesium acetate, 0.1 mM EDTA, 6 mM 2-mercaptoethanol, 2.5 mM spermine, 1.1 M sucrose). After centrifugation at 235,000 g for 19 h at 4 °C in a 45Ti rotor (Beckman), the supernatant was dialyzed against buffer A without spermine. This fraction (hereafter named S-100) was kept and stored at -80 °C for further use. The pellet containing ribosomes was dissolved in 4 ml of buffer A and loaded on top of sucrose gradients (10–30% sucrose in buffer A containing 2.5 mM magnesium acetate). After a 19 h centrifugation at 70,000 g (SW32.1 rotor, Beckman), the gradients were fractionated. Fractions containing either 50S or 30S subunits were pooled separately and the magnesium acetate concentration was increased to 18 mM. After centrifugation (220,000 g, 20 h), the pelleted subunits were dissolved in buffer A at a concentration of *ca* 3 μM and used for all complex preparations except that used for dataset 1. In this case, the subunits were prepared as described in refs. 56,57, in the presence of 100 mM $NH_4Cl$, and then dialyzed against buffer A.

### Production of initiation factors and tRNA

The gene encoding aIF1A from *S. solfataricus* was amplified from genomic DNA and cloned into pET3alpa[122] to produce a native version of the factor. BL21 Rosetta pLacIRare cells were transformed and cells were grown at 37 °C until $OD_{600nm}$ reached 2 to 3, then protein production was IPTG-induced for 5 hours. Cells were harvested and resuspended in lysis buffer (10 mM HEPES-NaOH pH 7.5, 500 mM NaCl, 3 mM β-mercaptoethanol) and lysed by sonication. After centrifugation, the supernatant was heated for 10 min at 80 °C. Precipitated material was removed by centrifugation and the supernatant was loaded onto an S-sepharose column equilibrated in buffer A plus 250 mM NaCl. aIF1A was eluted by applying an NaCl gradient (0.25–1 M) in buffer A. The recovered sample was then concentrated, flash-frozen and stored at −80 °C.

*E. coli* cells producing each subunit of the *S. solfataricus* aIF2 heterotrimer were mixed and the factor was purified as described[76]. A tagged version was also produced and used as indicated below. In this case, N-terminally His-tagged versions of the α and β subunits were used.

tRNA$_f^{Met}$ A1-U72, a mimic of archaeal initiator tRNA[57,123], hereafter called tRNA$_i^{Met}$, was produced in *E. coli* from a cloned gene, purified and aminoacylated using purified *E. coli* M547 methionyl-tRNA synthetase as described[124]. Met-tRNA$_f^{Met}$ was shown previously to bind Ss-aIF2 with a Kd of 1.5 nM[76].

### Preparation of total tRNA extracts from *S. solfataricus* and in vitro translation assay

A crude tRNA extract was prepared using standard protocols as described[124–126]. Briefly, 40 mL of the S-100 fraction were mixed with 40 mL of phenol saturated in 10 mM Tris-HCl pH 7.4, 10 mM $MgCl_2$. After centrifugation for 30 min at 15,000 g, the soluble fraction was precipitated by adding 0.6 volume of isopropanol and sodium acetate (150 mM final concentration). After centrifugation, the tRNA preparation was dissolved in 10 mM Glycine buffer pH 9.0 and incubated for 2 hours at 37 °C for tRNA deacylation. tRNAs were then ethanol precipitated and redissolved in buffer A without spermine. tRNA aminoacylation was performed in buffer A supplemented with 1 mM ATP in the presence of 20 μM [³H]-phenylalanine (specific activity 136 dpm/picomole, Perkin Elmer) and either purified *E. coli* PheRS or an aliquot of the S-100 fraction. Aminoacylation was observed in the presence of *E. coli* PheRS (1 μM in the assay) at 25 °C or with the S-100 (5-fold dilution in the assay) at 60 °C. No aminoacylation was observed with the S-100 when the reaction was incubated at 25 °C. The concentration of total RNA in the assay was estimated from $A_{260}$ to be 140 μM with approximately 1.4 μM of aminoacylatable tRNA$^{Phe}$.

In vitro poly-U translation conditions were adapted from[71,72]. 100 μL assays in buffer A contained 700 picomoles of tRNA extract, 20 μL S-100 fraction, 3.3 μL [³H]-phenylalanine (136 dpm/picomole), 2 mM GTP, 1.3 mM spermine, 4 μL of 10 mg/mL poly-U and 120 picomoles of 30S and 50S. The mixture was incubated at 60 °C for various times and then precipitated using 5 % trichloroacetic acid. Reaction mixtures were then heated at 90 °C for 15 minutes to hydrolyze aminoacyl tRNA[127] and finally the radioactivity incorporated in synthesized poly-Phe was recovered after filtration and counted. Results are shown in Supplementary fig. 1.

### In vitro transcription and synthetic RNAs

In vitro run-off transcription from linearized plasmid or from annealed oligonucleotides was done using T7 RNA polymerase essentially as described[75]. Transcripts were purified by ion exchange chromatography on a Mono-Q column (0.5 cm×5 cm, Cytiva). Other short RNAs were chemically synthesized by Dharmacon (Lafayette, USA) or Eurogentec (Seraing, Belgium). Supplementary Table 1 summarizes the sequences and origin of the mRNAs used in this study. Ss-MAP lmRNA starts with an A nucleotide which is not optimal for transcription

initiation. In this case, we used a Hammerhead ribozyme-based strategy[128]. Briefly, the ribozyme sequence was introduced between a strong transcription start site and the sequence coding for the mRNA. After transcription, the ribozyme was autocatalytically removed yielding the mRNA starting on AUG with a 5'-OH extremity.

## Toeprinting analyzes

Toeprinting experiments were performed essentially as described[75,129]. 30S complexes were assembled on mRNAs and were analyzed by primer extension using AMV reverse transcriptase (Promega). A typical mixture for 20 reactions was made as follows. Six picomoles of 5'-labeled (Dye-682 fluorophore, Eurofins) primer were annealed to 2.4 pmol of mRNA in 40 μl of annealing buffer (60 mM $NH_4Cl$, 10 mM HEPES pH 7.5, 7 mM 2-mercaptoethanol), in the presence of 40 units of RNase inhibitor (Invitrogen). After heating 3 min at 60 °C, the reaction was cooled to room temperature and magnesium acetate was added to reach a concentration of 60 mM. A premix (90 μl) was prepared by adding, to 40 μl of the above mixture, 20 μl of 30S subunits at a concentration of 1 μM beforehand activated 30 min at 65 °C, 10 μl of dNTPs (3.75 mM each), 10 μl GDPNP 5 mM and 120 units RNase inhibitor. After a 5 min incubation at 51 °C, the premix was distributed as 4.5 μl aliquots into 4 μl mixtures containing factors and Met-tRNA$_i^{Met}$ A$_1$-U$_{72}$ as desired (initiation factors were prediluted to 100 μM in buffer adapted to each factor and Met-tRNA was prediluted to 100 μM in water). Each tube was incubated 10 min at 51 °C, before addition of 1 μl of AMV Reverse Transcriptase (2 units/μl) and further incubated 15 min at 51 °C. The reactions were quenched by adding 3 μl of stop solution containing 95% formamide and blue dextran. A typical reaction mixture (9.5 μl) contained 0.12 pmol of mRNA (12.6 nM), 1 pmol of 30S (105 nM), 5 pmol of initiation factors (0.5 μM) and 10 pmol of Met-tRNA$_i^{Met}$ (1.05 μM) (Fig. 1a, see model-SD, Ss-EF1A-like and Ss-aIF2β lmRNAs). With Ss-MAP lmRNA, that has a 5'-OH extremity, 100 pmol of initiation factors and Met-tRNA were used (Fig. 1a).

When indicated, mRNAs were dephosphorylated using thermosensitive alkaline phosphatase FastAP (Thermo Fisher Scientific). Rephosphorylation was performed using T4-polynucleotide kinase (Thermo Fisher Scientific). mRNAs were quantified on denaturing acrylamide gels before use in toeprinting experiments (Fig. 1b).

For toeprinting experiments conducted in the presence of Ss-eS26, the protein was added after mixing mRNA and 30S, with a range of concentrations from 0 to 1.2 μM (Fig. 1c). The mixture was then heated for 5 minutes at 51 °C before continuing the protocol as described above for the typical conditions. When indicated (Supplementary fig. 22), 30S were preincubated 5 minutes at 51 °C before adding mRNA and continuing the protocol.

Reverse transcripts were analyzed on a Licor 4200 DNA sequencer. Sequencing reactions made with the corresponding plasmidic DNA and the RT primer were loaded in order to identify the toeprinting positions. Images were processed using the ImageJ software for quantitative analysis[130]. The toeprinting signal corresponding to the major reverse transcriptase arrest (noted +16 or +17 in Supplementary figs. 3, 17 and 20) was quantified (T) as well as the read-through signal (RT bands in Supplementary figs. 3, 17 and 20). Figure 1 shows the percentage of T relative to T + RT. The numbers of repetitions are indicated in Figure legends. A repetition corresponds to a full toeprinting experiment (experimental unit), as described in this subsection.

## Production and purification of *S. solfataricus* eS26

*S. solfataricus* cells were directly used for amplifying the eS26 gene with Pfu-cloner DNA polymerase. The PCR product was inserted between the NdeI and NotI sites of the pET21a plasmid, resulting in the expression of a protein carrying a (His)$_6$ tag in C-terminal position. Finally, site-directed mutagenesis was used to introduce a thrombin

cleavage site between the eS26 and the His-Tag coding sequences to allow His-tag removal.

Ss-eS26 was expressed in *E. coli* Rosetta cells in 1 L of auto-inducible Terrific Broth medium (ForMedium AIMTB0260) supplemented with ampicillin and chloramphenicol. At mid-exponential phase, the culture was transferred from 37 °C to 20 °C for 16 hours. Harvested cells were sonicated in buffer B (10 mM HEPES pH=7,5; 1 M NaCl; 3 mM β-mercaptoethanol) supplemented with PMSF and benzamidine. The extract was clarified by centrifugation (30 min, 20 000 g, 4 °C) and Ss-eS26 was purified via Talon Metal affinity chromatography (Clontech). (His)$_6$-tag was removed by thrombin digestion in buffer B containing 250 mM NaCl. Uncleaved molecules were removed by using Talon Metal affinity chromatography. The preparation was finally polished by size-exclusion chromatography on a Superdex 75 10/300 column (GE-Healthcare). An aliquot of this preparation was sent to the ProteoGenix company (Schiltigheim, France) in order to derive and affinity purify rabbit polyclonal antibodies.

## Western blotting with eS26 antibodies

Proteins were loaded on SDS-PAGE and were transferred on a nitro-cellulose membrane during 1 hr at 100 V with ice pocket in an electrophoresis chamber. The membrane was incubated in nonfat dry milk (3% in PBS) for 30 min at room temperature. Blocking solution was replaced by 5 mL of 1/500 diluted anti-eS26 antibodies in 3% nonfat dry milk-PBS for at least one hour incubation. Membrane was washed 3 times with PBS-T (PBS with 0.1% of tween 20), incubated with 1/5000 diluted peroxidase-labeled secondary antibodies (Sigma-Aldrich) for 30 min. The membrane was finally revealed by chemiluminescence (Lumilight Plus system, Roche) using a Chemidoc apparatus (BioRad, USA).

## Sequence analyzes

Sequences in the 5' untranslated regions of *S. solfataricus* genes were analyzed as follows. mRNAs were considered as leadered if the 5'-UTR was at least 11 nucleotides according to the data in ref. 48. Exact occurrences of GGAGG or GGTGG sequences were searched for in the upstream regions (30 nucleotides before the annotated start codon). The sequence was considered as an SD signal if the 5' G base of the searched sequence was located between the -18 and -9 positions of the annotated start codon or of another in-frame start codon.

Oligonucleotides used in this study are listed in Supplementary Table 4.

## Phylogenetic distributions of aS33, aS34 and of eS6 domain 2

Homologs of *S. solfataricus* aS33 and aS34 sequences were searched for in the non-redundant protein sequences database at NCBI using BLASTP[131]. The taxonomy of species containing homologs was then examined with the NCBI Taxonomy tool. With the exception of one unidentified "Thermoproteota archaeon", all species belonged to the Thermoprotei class, and more precisely to the Acidilobales, Sulfolobales, Desulfurococcales and Fervidicoccales families. In contrast, Thermoproteales and Thermofilales were not represented. We verified that searches restricted to Thermoproteales and Thermofilales failed to retrieve any convincing homolog. This phylogenetic distributions of aS33 and aS34 was confirmed by HMMER searches[132] on the EMBL-EBI server.

Similarly, the sequence of *S. solfataricus* eS6-domain 2 (118 amino acids) was searched for using BLASTP. Again, only sequences from Acidilobales, Sulfolobales, Desulfurococcales and Fervidicoccales were retrieved. Accordingly, analysis of eS6 proteins from Thermoproteales or Thermofilales showed the absence of eS6-domain2.

## Preparation of initiation complexes for cryo-EM analyzes

*S. solfataricus* 30S subunits were heated for 30 min at 65 °C before use. For DS1, subunits in buffer A were mixed with a 5-fold excess of Ss-

aIF1A and of the short version of the model-SD-mRNA. The mixture was incubated for 3 min at 65 °C. Pre-formed ternary complex aIF2:GDPNP:Met-tRNA$_i^{Met}$A$_1$-U$_{72}$ (2-fold excess relative to ribosomal subunits) was added and the mixture heated 30 sec at 65 °C. The complex formed was then purified by size exclusion chromatography using an Agilent Bio SEC-5 HPLC column (1000 Å). The fraction corresponding to the top peak was recovered, concentrated with a Centricon 30 and used for grid preparation. Cryo-EM images were collected as detailed in Supplementary Table 2.

The initiation complexes used for DS2, DS3 and DS4 were prepared using a similar strategy, except that His-tagged versions of Ss-aIF2α and Ss-aIF2β were used. Moreover a 10-fold excess (relative to 30S) of Ss-eS26 was added in the mixture. The initiation complexes were then purified using Talon (Clontech) affinity chromatography as described[57]. The recovered complex was dialyzed against buffer A and concentrated to 100 nM using Centricon 30, and then stored at -80 °C after flash freezing in liquid nitrogen. The presence of the components was confirmed by SDS-PAGE (Supplementary fig. 4c).

### Cryo-EM analysis

Before spotting onto the grids, an excess (3-fold relative to 30S) of aIF1A, TC plus 5 μM GDPNP was added for DS1. For DS2, DS3 and DS4 we also added an excess of Ss-eS26 (5-fold relative to 30S). For DS4, we also added a 50x excess of aIF1A in the preparation. IC were heated 1 min at 50 °C and immediately used for grid preparations.

Complexes (3.4 μL) were spotted onto grids (Quantifoil Copper 300 mesh with an additional 2 nm continuous carbon layer) at 20 °C and 90% humidity for 10 s. The sample was vitrified by plunging into liquid ethane at −182 °C, after 1.2 s blotting using a Leica EM-GP plunger. Conditions were optimized using a Titan Themis cryo-microscope (ThermoFisher) at the CIMEX facility of Ecole Polytechnique. Final datasets (Supplementary Table 2) were collected on Titan Krios or Glacios cryo-microscopes (ThermoFisher).

### Cryo-EM data processing

DS1, DS2 and DS3 were processed in RELION[133]. Frames were aligned using 5×5 patches with 1 e-/Å$^2$/frame dose weighting. CTF was estimated using gCTF[134] on the dose-weighted micrographs. Micrographs with gross surface contamination or obvious gCTF output-metrics outliers were discarded. Particles were extracted from clean sorted micrographs (resolution, astigmatism, defocus) and sorted by 2D-classification followed by 3D-classification. Sorting the h44-up and -down conformations was done using focused 3D classification procedures. The quality of the cryo-EM maps was further improved by using post-processing and per particle CTF refinement.

Data processing of DS4 was conducted in Cryosparc 4[135]. Initial 2D templates were generated via blob picking and used to train a Topaz model[136]. Topaz picks were then used to generate three initial models, which were refined using heterogeneous refinement. 3D classifications yielded three maps that differ by the presence or absence of a tRNA at the P site and by h44 helix conformation. All particles with a tRNA at the P-site were grouped and refined to produce the final 3.7 Å resolution map (Supplementary fig. 5).

All structural figures were drawn using ChimeraX[137].

### Mass spectrometry analysis of 30S subunits

30S preparations were diluted in 8 M urea, 100 mM Tris HCl pH 8.5 to obtain a final urea concentration of 6 M. Proteins were reduced using 5 mM Tris(2-carboxyethyl)phosphine for 30 min at room temperature. Alkylation of the reduced disulfide bridges was performed using 10 mM iodoacetamide for 30 min at room temperature in the dark. Proteins were then digested in two steps, first with 500 ng r-LysC Mass Spec Grade (Promega) for 4 h at 30 °C and then samples were diluted below 2 M urea with 100 mM Tris HCl pH 8.5 and 500 ng Sequencing Grade Modified Trypsin was added for the second digestion overnight

at 37 °C. Proteolysis was stopped by adding formic acid (FA) at a final concentration of 5%. The resulting peptides were desalted on Stage Tip[138] prepared with Empore 3 M C$_{18}$ material (Fisher Scientific). Peptides were eluted using 50 % acetonitrile (ACN), 0.1 % formic acid (FA). Peptides were concentrated to dryness and resuspended in 2 % ACN/ 0.1 % FA just prior to LC-MS injection.

LC-MS/MS analysis ($n = 1$) was performed on a Q Exactive™ Plus Mass Spectrometer (Thermo Fisher Scientific) coupled with a Proxeon EASY-nLC 1200 (Thermo Fisher Scientific). One μg of peptides was injected onto a home-made 30 cm C$_{18}$ column (1.9 μm particles, 100 Å pore size, ReproSil-Pur Basic C18, Dr. Maisch GmbH, Ammerbuch-Entringen, Germany). Column equilibration and peptide loading were done at 900 bars in buffer A (0.1 % FA). Peptides were separated with a multi-step gradient from 3 to 6 % buffer B (80% ACN, 0.1% FA) in 5 min, 6 to 31 % buffer B in 80 min, 31 to 62 % buffer B in 20 min at a flow rate of 250 nL/min. Column temperature was set to 60 °C. MS data were acquired using Xcalibur software using a data-dependent method. MS scans were acquired at a resolution of 70,000 and MS/MS scans (fixed first mass 100 m/z) at a resolution of 17,500. The AGC target and maximum injection time for the survey scans and the MS/MS scans were set to 3E$^6$, 20 ms and 1E$^6$, 60 ms respectively. An automatic selection of the 10 most intense precursor ions was activated (Top 10) with a 30 s dynamic exclusion. The isolation window was set to 1.6 m/z and normalized collision energy fixed to 27 for HCD fragmentation. We used an underfill ratio of 1.0 % corresponding to an intensity threshold of 1.7E$^5$. Unassigned precursor ion charge states as well as 1, 7, 8 and > 8 charged states were rejected and peptide match was disabled.

### Protein identification

Acquired Raw data were analyzed using MaxQuant software version 2.1.1.0[139] using the Andromeda search engine[140,141]. The MS/MS spectra were searched against the *Saccharolobus solfataricus* Uniprot reference proteome database (2980 entries), the 30S ribosomal protein eS26 (A0A0E3MCW1) and the D0KTI0 sequences.

All searches were performed with oxidation of methionine and protein N-terminal acetylation as variable modifications and cysteine carbamidomethylation as fixed modification. Trypsin was selected as protease allowing for up to two missed cleavages. The minimum peptide length was set to 5 amino acids and the peptide mass was limited to a maximum of 8,000 Da. One peptide unique to the protein group was required for the protein identification. The main search peptide tolerance was set to 4.5 ppm and to 20 ppm for the MS/MS match tolerance. Second peptides were enabled to identify co-fragmentation events and match between runs option was selected with a match time window of 0.7 min over an alignment time window of 20 min. The false discovery rate (FDR) for peptide and protein identification was set to 0.01.

### Mass spectrometry analysis of hydrolyzed 16S rRNA

Fully hydrolyzed 16S rRNA suitable for mass spectrometry analysis was prepared as described[142]. Samples were then diluted in 0.1 % formic acid (FA) prior to analysis. Chromatographic grade solvents (99.99% purity), acetonitrile (ACN) and formic acid, were purchased from Sigma Aldrich. Liquid chromatography/ high-resolution mass spectrometry (LC-HRMS) analyzes were performed with the timsTOF mass spectrometer coupled with an Elute HPLC system (Bruker Daltonics, Bremen, Germany). The sample (10 μL, 2.5 μg digested 16S rRNA) was injected and separated on an Atlantis T3 column (3 μm, 150 × 2.1 mm; Waters, Saint Quentin, France). The effluent was introduced at a flow rate of 0.2 mL.min$^{-1}$ into the interface with a gradient increasing from 10% of solvent B to 50% in 6 min to achieve 70% at 8 min (A: water with 0.1% FA; B: methanol with 0.1% FA). From 8 min to 12 min, the percentage of solvent increased up to 90% of B. The flow was then set at 10% of B for the last 6 min. Electrospray ionization was operated in the positive ion mode. Capillary and end plate voltages were set at −4.5 kV

and −0.5 kV, respectively. Nitrogen was used as the nebulizer and drying gas at 2 bar and 8 L/min, respectively, with a drying temperature of 220 °C. In MS/MS experiments, the precursor ion was selected with an isolation window of 1 Da and the collision-induced dissociation was performed using collision energies (Ecol) ranging from 7 eV to 25 eV. Tuning mix (Agilent, France) was used for calibration. The elemental compositions of all ions were determined with the instrument software Data analysis, the precision of mass measurement was better than 3 ppm. All nucleosides have been characterized by both molecular MH+ and fragment ions BH+ (Supplementary Table 5). The only exception was $m_1acp^3\psi$ for which only the MH+ ion was identified.

### Reporting summary

Further information on research design is available in the Nature Portfolio Reporting Summary linked to this article.

## Data availability

The atomic models and cryo-EM maps generated in this study have been deposited in the Protein Data Bank and EMDB under accession code as follows, 30S-HR: 9FHL, EMD 50445; DS2-IC2-down 9FRA, EMD 50709; DS2-IC2-up 9FSF, EMD 50727; DS1-IC2-down, 9FRK, EMD 50716; DS1-IC2-up, 9FRL, EMD 50717; DS3-IC2-down 9FY0, EMD 50854; DS3-IC2-up 9FS6, EMD 50724; DS4-IC2-down 9FS8, EMD 50725. The mass spectrometry proteomics data have been deposited to the ProteomeXchange Consortium via the PRIDE partner repository with the dataset identifier PXD053103. Sequence alignments generated in this study are provided in the supplementary information and source data file. Source data are provided with this paper.

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

## Acknowledgements

This work was supported by grants from the Centre National de la Recherche Scientifique and Ecole polytechnique to Unité Mixte de Recherche n°7654, by a grant from the Agence Nationale de la Recherche (ANR-17-CE11–0037; TREMTI) and served as preliminary results for grant ANR-24-CE11-4225 (LmRNA). Cryo-EM data benefited from access to CM01 beamline (ESRF, Grenoble), the Nanoimaging facility (Institut Pasteur, Paris, created with the help of ANR-11-EQPX-008 grant) and the Interdisciplinary Center for Electron Microscopy of Ecole Polytechnique (CIMEX). We thank the staff at these facilities, and in particular Isai Kandiah and Daouda Traore (ESRF), Stéphane Tachon (Institut Pasteur), Eric Larquet and Kassiogé Dembélé (CIMEX). We thank David Mignon for skillful computer assistance, Michel Fromant for the gift of *S. solfataricus* cells and Antoine Mechulam for writing useful computer scripts. We thank Sebastien Ferreira-Cerca for helpful discussions.

## Author contributions

E.S. and Y.M. designed research. G.B. and C.L.S. performed the biochemical experiments and initiation complexes preparation. G.B., P.D.C., C.M., Y.M., and E.S. did cryo-EM analysis, model building, and data analysis. S.B., M.D., and J.C.R. performed and analyzed the mass-spectrometry experiments. TG performed explicit water molecular dynamics simulation. E.S. drafted the manuscript with input from all authors. All authors reviewed and edited the manuscript.

## Competing interests

The authors declare no competing interest.
