## [Transparent Peer Review file · Nature Communications]

Structures of *Saccharolobus solfataricus* Initiation Complexes with Leaderless mRNAs Highlight Archaeal Features and Eukaryotic Proximity

Corresponding Author: Dr Emmanuelle SCHMITT

Version 1:

Reviewer comments:

Reviewer #1

(Remarks to the Author)

I have read this manuscript by Gabrielle Bourgeois and colleagues with great interest and pleasure. This work provides insights into one of the most enigmatic evolutionary events that gave rise to eukaryotic species. Through their structural studies of ribosomes from the closest relatives of eukaryotic cells, the archaeon *Saccharolobus solfataricus*, the authors of this study shed light on the evolutionary process that allowed organisms to transition from the bacterial mode of mRNA translation, which depends on the presence of Shine-Dalgarno sequences in mRNA, to the eukaryotic mode of mRNA translation, which relies on the recognition of the 5'-cap of mRNA.

In recent years, the archaeal domain of life has garnered significant attention due to its unique position between bacteria and eukaryotes. As a result, many riddles regarding the origin of eukaryotes can be hidden within the structure of one of the most complex and conserved molecular assemblies of the archaeal cells—their ribosomes. Specifically, it is widely recognized in the field that archaeal ribosomes serve as a "structural hybrid" between eukaryotic and bacterial ribosomes. Similarly to eukaryotes, archaea possess ribosomal proteins that are shared by many archaea and eukaryotes but are absent in bacteria. However, unlike eukaryotes and akin to bacteria, the 16S rRNA of archaeal ribosomes features the characteristic mRNA-binding element known as the anti-Shine Dalgarno sequence. This suggests that archaeal ribosomes may employ mechanisms of mRNA translation shared by both eukaryotes and bacteria.

In this study, the authors set up an experimental system to test this hypothesis by capturing archaeal ribosomes in the process of initiating protein synthesis. This allowed the authors to observe how archaeal ribosomes utilize their anti-Shine-Dalgarno sequences. I find it so fascinating that their model organism appears to use two alternative mechanisms of protein synthesis, including both the bacterial and eukaryotic modes to initiate translation of mRNA. What's more, the authors show that the switch between these two modes is likely triggered by the ribosome association with protein eS26. This finding potentially means that protein eS26, which is currently viewed as a bona fide ribosomal protein in eukaryotes, may act more like a translation regulator in archaeal ribosomes—by sequestering the anti-Shine-Dalgarno sequence in the archaeal 16S rRNA.

I agree with the authors' conclusion that their observed structure indeed provides insight into how living organisms transitioned from the bacteria-type translation of messenger RNAs to the type observed in eukaryotes. They have described one likely mechanism of this evolutionary transition, mediated by the acquisition of the ribosomal protein eS26, which eventually helped give rise to eukaryotic life on our planet.

I would be happy to see this work published in Nature Communications, but I suggest the following corrections to make this manuscript more rigorous and accessible to a general reader.

Major comment:

1. The finding of eS26 only in one of the two functional complexes with mRNA is remarkable and deserves a better explanation and possibly an additional analysis.
 - a. I wonder if the authors could estimate the occupancy of eS26 in vacant 30S subunits that were used for their study. They

provide the mass spectrometry evidence (Figure S2) but this evidence is not strictly quantitative. As a result, we do not know what fraction of the 30S subunits (or 70S ribosomes) contain eS26 in this archaeal organism. I wonder if the authors could address this through focused particle classification or recollecting data on the vacant 30S ribosomes/70S ribosomes?

I find this question key to their study because their fascinating data imply two alternative scenarios: they either mean that archaeal cells bear two populations of functionally specialized ribosomes (one for leaderless mRNAs and one for SD-type mRNAs) or that, alternatively, eS26 acts as a translation factor that binds ribosomes in a reversible manner to facilitate the recruitment of leaderless mRNAs. I feel that the authors could make a much stronger conceptual statement by providing these additional data/analyses. However, I am also open to their verbal response because I understand that some experiments can be very hard to execute.

b. Their toeprinting assays suggest that eS26 is not an integral/stoichiometric component of the ribosome but instead acts as a factor that stimulates the translation of leaderless mRNAs in archaea. I wonder if the authors could test whether eS26 binding to the ribosome is reversible or biochemically stable. In other words, how would their toeprinting results change if, instead of adding eS26 to the reaction mixture, they first treated the 30S subunit with excessive amounts of eS26, then removed the non-bound eS26, and only then used the obtained 30S subunits for experiments? Does this experiment make sense?

2. The authors could describe the newly identified archaeal ribosomal proteins a little bit better.

a. The authors' analysis (e.g., Figures 3B and 4) shows that protein eS6 is conserved only in Thermoprotei, but not in other Archaea, including the closest eukaryotic relative, Asgard. How do the authors explain the origin of eS6 in eukaryotes and its absence in Asgard?

b. I would expect eS6 to be labeled in Figure 1.

c. In Figure 4, could the authors also show the structures of proteins aS33, aS34 and eS6—so that a reader can judge about their size and three-dimensional structure? These are key discoveries of the paper and they are barely shown in their current figures.

d. Are the genes for proteins aS33, aS34, and eS6 organized in a single operon?

e. The authors should consider explicitly stating that the eS6-domain-containing protein identified in this study has the same location in the archaeal ribosome as protein eS6 in eukaryotic ribosomes. They could show this, for instance, in their Figure 3 by adding panel C.

Minor comments:

1. Archaeal species are known to greatly differ in the content of ribosomal proteins, often lacking one or a few proteins that are present in other archaeal species (<https://pubmed.ncbi.nlm.nih.gov/12490706/>). Because the structure of *S. solfataricus* small subunit appears to be the most complete and even includes additional proteins, I suggest the authors label all individual ribosomal proteins in the ribosome structure (instead of labeling just the archaeal-specific proteins they have identified in their study). This could help make this work an independent reference for everyone interested in the structure and evolution of the archaeal translation machinery.

2. I would consider renaming the following sections for clarity: "Protein specificities" to "Protein features" and "Specificities of the small ribosomal subunit from *S. solfataricus*" to "Specific features of the small ribosomal subunit from *S. solfataricus*".

3. Consider adding essential detail when referring to the structural features of ribosomal proteins. For example, instead of writing "In addition, in eukaryotic initiation complexes, eIF1A is bound to the A site and contacts the N-terminal tail of eS30", consider specifying the residues: "In addition, in eukaryotic initiation complexes, eIF1A is bound to the A site and contacts the N-terminal tail of eS30, including residues 2-15". Also, make sure you use consistent naming for amino acids (e.g. currently I see "Proline 2" and "P in position 2", which is a bit confusing, especially when you use the same "P" to refer to the P site of the ribosome).

To conclude, I want to restate that this work presents genuinely novel knowledge about an important biological problem. It is highly intellectually stimulating and I will be happy to support its publication in Nature Communications.

Reviewer #2

(Remarks to the Author)

This paper describes a detailed study by cryo-EM and other techniques of the structure of translation initiation complexes of the crenarchaeon *Saccharolobus* (formerly *Sulfolobus*) *solfataricus*. The ribosomes of this organism have previously been shown to share structural and functional features with those of eukaryota. Many studies have been focused on the translational initiation step, which differs markedly between bacteria on one side, and eukaryotes and archaea on the other. Elucidating the details of translation initiation in archaea, especially those belonging to the TACK branch which is the closest to eukaryotes, is very important in order to understand the evolutionary history of translation, and also to unravel the relationship between archaea and eukaryotes.

The most notable results of the present study are the detailed description of translation initiation complexes with both leaderless and leadered mRNAs, including the elucidating of the role of the SD sequences (where present) and of the

terminal triphosphate group in leaderless messages. Particularly notable is the discovery of two novel ribosomal proteins having a role in these processes, and the detailed elucidation of the position and function of three proteins shared by the TACK archaea and the eukaryotes.

The work is well done, well conceived and technically sound. It is so rich in valuable information that it is sure to become a main reference paper for those involved in the fields of translation and evolution of translation. Among many novel and intriguing findings, I found especially interesting the description of alternative conformational states for helix 44 in the small ribosomal subunit (which may finally help to explain the long-observed and still unexplained instability of monomeric ribosomes in *S. solfataricus* and similar crenarchaea). Moreover, the paper is very well and clearly written, the methodology information is exhaustive, and the bibliography more than adequate.

Reviewer #3

(Remarks to the Author)

The manuscript by Bourgeois et al. reports on the functional and structural analysis of archaeal ribosomes and their complexes with either leaderless mRNA or mRNAs comprising a Shine-Dalgarno (SD) sequence. The authors report on a number of interesting novel findings regarding the specific composition and structure of the 30S ribosomal subunit of one particular archaeal species, *S. solfataricus*, which considering recent / ongoing phylogenetic studies is an interesting approach to also understand general evolutionary aspects (which are often based on ribosomal RNA as ribosomes are fundamental cellular enzymes). Two new archaea-specific ribosomal proteins were discovered (aS33 and aS34, validated by mass spectrometry and visible in the cryo electron microscopy structures), in addition to archaea-specific versions of eS6, eS26 and other proteins. Ribosomal protein aS33 and aS34 appear to co-occur in archaeal sub-species, giving insights into co-evolution. Ribosomal protein eS26 is shown to compete with mRNA and may serve as a trigger to switch between SD-leadered and leaderless mRNA 30S translation initiation complexes. Leaderless mRNA is found to comprise a 5'-triphosphate moiety that contributes to stabilization of 30S/mRNA complexes as observed in bacterial ribosomes, which is corroborated by a comparison of tri-, de- and mono-phosphorylated versions of the mRNA using toeprinting analysis. There are also some interesting features on chemical modifications of the rRNA, which compared to bacterial and eukaryotic/human ribosomes is interesting to compare with. The 3' end of the 16S rRNA is shown to be processed, which is a new finding too regarding 16S maturation in archaea.

Taken together, this work puts together many complementary tools combining functional and structural analysis, which provides many novel insights into archaea-specific ribosomes and the molecular mechanism of translation initiation. The manuscript is well written and is complemented by a series of supplementary data. Some detailed points and minor corrections are suggested below for a final (minor) revision.

Detailed points:

- abstract: maybe add as a conclusion (to clarify) that eS26 favors leaderless mRNA binding
- introduction: "no bacterial-type proteins are present in the archaeal ribosome": maybe reformulate because the universal ribosomal proteins are also bacterial
- introduce SD abbreviation
- add "helix" to h44 to say "helix h44" (in several places); same for helix h16
- which implication could have the 2 conformations of helix h44? Is there any factor-dependence on this? Or is this simply a feature reminiscent of the absence of the 50S ribosomal subunit?
- usage of high Mg acetate concentration: Mg²⁺ is known to stabilize RNA complexes, but may also lead to specificity loss in interactions
- cryo-EM processing: classifications using masks refer to focused classifications (and refinements), see a recent paper in JSB and references therein
- chemical modifications of the rRNA: could be compared with other bacterial and eukaryotic species, including human (for the few conserved across species; see for example a recent analysis in NSMB 2024, Suppl Data Table which comprises human and other eukaryotic species)
- m1acp3PSU modification found in eukaryotes, indeed e.g. in human: see PDB IDs 6QZP and 8QOI (and corresponding references)
- chemical modification of A1475/1476: usually this site has two dimethyl modifications (2 methyl's on each adenine), is there only one modified here? see also Suppl. Fig. 9, which seems to show only one modification, as suggested also from primer extension analysis (Suppl. Fig. 10); both in bacteria and human etc. this is a double-modification site
- comparisons with human ribosomes: it could be useful to use the latest high-resolution work for this (NSMB 2024); in particular, the interactions described for eS26, eS25, eS30 etc. are visible in there also: eS26 is well visible and is in part close to the mRNA channel; eS25 is there and its N-ter becomes ordered upon tRNA presence; eS30 N-ter visible from 5. residue onwards
- proposed π -stacking between Proline and glycosidic bond: from a chemistry point of view there is no aromatic system involved, hence this sounds more like a van der Waals contact; to be checked
- 2 N-terminal glycine residues of eS25: what are the specific molecular interactions of these residues, backbone hydrogen bonding possibly? Maybe add figure panel in Suppl. Fig. 12 to show the detailed interactions
- page 11: clarify which protein gets which name (aS34 and aS33)
- co-occurrence of ribosomal protein aS33 and aS34: is there any evidence for co-evolution, also with eS6, from sequence alignments? But on the other side they seem to be far from each other on the structure. Would AlphaFold be able to predict these 2 proteins?
- page 12: stabilization of codon-anticodon duplex with uS19: has been seen also in human, see Bhaskar et al., 2020
- "potential map": cryo-EM maps are indeed Coulomb potential or electrostatic potential maps, but this seems to add little to

the discussion in the text (unless there would be charge effects to be considered); maybe simply refer to “cryo-EM maps”

- binding of 2 mRNA molecules: a) if so, what could this involve functionally? b) to be on the safe side, are these 2 molecules clearly identified individually from the sequence visible in the cryo-EM map, i.e. are the same sequence patches observed twice (e.g. no overlap of 2 fragments visible)?
- page 16, position of eS26 in yeast and human: also in 6QZP and 8QOI (and corresponding references); also in Fig. 5B legend; same for eS30, also in Suppl. Fig. 11A and 12B legends; in general, references should better be added to PDB IDs
- references to resolution estimation from Fourier shell correlation (FSC) should normally comprise 1) van Heel M., Keegstra W., Schutter W. G. & van Bruggen E. F. J. Arthropod hemocyanin studied by image analysis. *Life Chem Rep Suppl* 69–73 (1982). 2) Saxton, W. O. & Baumeister, W. The correlation averaging of a regularly arranged bacterial cell envelope protein. *J. Microsc.* 127, 127–138 (1982). 3) Rosenthal, P. B. & Henderson, R. Optimal determination of particle orientation, absolute hand, and contrast loss in single-particle electron cryomicroscopy. *J. Mol. Biol.* 333, 721–745 (2003).
- Suppl. Fig. 7, update of secondary structures according to the experimental structure: is it maybe possible to feedback this information into the database to have correct annotations for everyone using the database?
- it could be useful to have a validation report for the refined coordinates & pdb deposition, for example to check for clashes in the atomic model etc.
- generic point: cryo-EM maps, half maps, masks etc. and fully refined atomic models should be deposited in the EMDB and PDB; deposition of representative data onto the EMPIAR data base could be considered also
- figures look good; suggestion for Fig. 8: maybe transfer to Suppl. Figs?

Thank you for the opportunity to review this work.
Bruno Klaholz

Version 2:

Reviewer comments:

Reviewer #1

(Remarks to the Author)

Thank you very much for addressing all my concerns and implementing useful advice. I am fully satisfied with this version of the manuscript.

Reviewer #2

(Remarks to the Author)

I already gave a very positive review of this manuscript, for the reasons detailed in my previous assessment. I have not much to add to that report. This is a very good paper that will be extremely useful to investigators working in the field, therefore it surely deserves publication.

Reviewer #3

(Remarks to the Author)

Thank you for the revised version of the manuscript.

The authors have taken into account all suggestions and specified certain points in the manuscript where asked for.

No other comments. I would recommend publication.

Thank for the opportunity to review this work.

Bruno Klaholz

We thank the reviewers for their positive and kind comments on our work. We are grateful for their meticulous work that helped us to add new experiments to improve our manuscript and conclusions. We have carefully considered their comments as detailed below.

Reviewer #1:

I have read this manuscript by Gabrielle Bourgeois and colleagues with great interest and pleasure. This work provides insights into one of the most enigmatic evolutionary events that gave rise to eukaryotic species. Through their structural studies of ribosomes from the closest relatives of eukaryotic cells, the archaeon *Saccharolobus solfataricus*, the authors of this study shed light on the evolutionary process that allowed organisms to transition from the bacterial mode of mRNA translation, which depends on the presence of Shine-Dalgarno sequences in mRNA, to the eukaryotic mode of mRNA translation, which relies on the recognition of the 5'-cap of mRNA.

In recent years, the archaeal domain of life has garnered significant attention due to its unique position between bacteria and eukaryotes. As a result, many riddles regarding the origin of eukaryotes can be hidden within the structure of one of the most complex and conserved molecular assemblies of the archaeal cells—their ribosomes. Specifically, it is widely recognized in the field that archaeal ribosomes serve as a "structural hybrid" between eukaryotic and bacterial ribosomes. Similarly to eukaryotes, archaea possess ribosomal proteins that are shared by many archaea and eukaryotes but are absent in bacteria. However, unlike eukaryotes and akin to bacteria, the 16S rRNA of archaeal ribosomes features the characteristic mRNA-binding element known as the anti-Shine Dalgarno sequence. This suggests that archaeal ribosomes may employ mechanisms of mRNA translation shared by both eukaryotes and bacteria.

In this study, the authors set up an experimental system to test this hypothesis by capturing archaeal ribosomes in the process of initiating protein synthesis. This allowed the authors to observe how archaeal ribosomes utilize their anti-Shine-Dalgarno sequences. I find it so fascinating that their model organism appears to use two alternative mechanisms of protein synthesis, including both the bacterial and eukaryotic modes to initiate translation of mRNA. What's more, the authors show that the switch between these two modes is likely triggered by the ribosome association with protein eS26. This finding potentially means that protein eS26, which is currently viewed as a bona fide ribosomal protein in eukaryotes, may act more like a translation regulator in archaeal ribosomes—by sequestering the anti-Shine-Dalgarno sequence in the archaeal 16S rRNA.

I agree with the authors' conclusion that their observed structure indeed provides insight into how living organisms transitioned from the bacteria-type translation of messenger RNAs to the type observed in eukaryotes. They have described one likely mechanism of this evolutionary transition, mediated by the acquisition of the ribosomal protein eS26, which eventually helped give rise to eukaryotic life on our planet.

I would be happy to see this work published in Nature Communications, but I suggest the following corrections to make this manuscript more rigorous and accessible to a general reader.

We thank the reviewer for his careful reading and appreciation.

Major comment:

1. The finding of eS26 only in one of the two functional complexes with mRNA is remarkable and deserves a better explanation and possibly an additional analysis.
 - a. I wonder if the authors could estimate the occupancy of eS26 in vacant 30S subunits that were used for their study. They provide the mass spectrometry evidence (Figure S2) but this evidence is not strictly quantitative. As a result, we do not know what fraction of the 30S subunits (or 70S ribosomes) contain eS26 in this archaeal organism. I wonder if the authors could address this through focused particle classification or recollecting data on the vacant 30S ribosomes/70S ribosomes?

I find this question key to their study because their fascinating data imply two alternative scenarios: they either mean that archaeal cells bear two populations of functionally specialized ribosomes (one for leaderless mRNAs and one for SD-type mRNAs) or that, alternatively, eS26 acts as a translation factor that binds ribosomes in a

reversible manner to facilitate the recruitment of leaderless mRNAs. I feel that the authors could make a much stronger conceptual statement by providing these additional data/analyses. However, I am also open to their verbal response because I understand that some experiments can be very hard to execute.

The stoichiometry of Ss-eS26 in vacant ribosomes is indeed a very important question to assess in vivo translation regulation. To address the point raised by the reviewer we first quantified Ss-eS26 in our 30S preparations using western blotting and antibodies directed against our purified Ss-eS26. By comparing the signal in the 30S with that of a range of eS26 concentrations we determined that our 30S preparations contain about 5% of eS26. A representative western blot experiment is now shown in Supplementary Figure 21a. In addition, we collected a cryo-EM dataset with our vacant 30S subunits. 4158 images were collected. 408,560 particles were used to calculate a 4.55 Å resolution cryo-EM map after alignment on the 30S bodies to account for head mobility. As shown in Supplementary Figure S21b, eS26 was not detected in the exit chamber even at low threshold values. We then performed extensive trials of 3D classifications, not-focused or focused on various regions encompassing the eS26 site. However, this did not allow us to identify an eS26-containing class. This is consistent with the low amount of eS26 deduced from the Western blot analysis. Moreover, because eS26 is in equilibrium with the 30S, the necessary dilution to 100 nM before spotting on grids likely contributed, by mass action, to diminish even more the fraction of bound eS26. These two new experiments are now described in the text at the end of the results section on page 16 and are further discussed in the discussion section on page 18. The methods section has also been updated to describe the western blot experiments.

b. Their toeprinting assays suggest that eS26 is not an integral/stoichiometric component of the ribosome but instead acts as a factor that stimulates the translation of leaderless mRNAs in archaea. I wonder if the authors could test whether eS26 binding to the ribosome is reversible or biochemically stable. In other words, how would their toeprinting results change if, instead of adding eS26 to the reaction mixture, they first treated the 30S subunit with excessive amounts of eS26, then removed the non-bound eS26, and only then used the obtained 30S subunits for experiments? Does this experiment make sense?

From the new experiments added as suggested by the reviewer, we now know that only a small fraction of purified Ss-30S subunit contains Ss-eS26. This fraction is much smaller than the proportion of l-mRNA in *S. solfataricus*. This is a strong indication that eS26 is reversibly bound to the Ss-30S and thus does not quantitatively co-purify with the 30S. When eS26 is added in excess with respect to the Ss-30S subunit concentration, we do see its binding as shown in DS3 and in toeprinting experiments (Figure 1c and Supplementary Figure 20a and b).

eS26 competes with leadered mRNAs for binding to 30S subunits. In our toeprinting experiments (Supplementary Figure 20) the concentration of 30S was 0.1 μM. eS26 was added as an initiation factor and concentrations greater than 0.3 μM were necessary to destabilize the initiation complexes formed on leadered mRNAs. This again argues in favor of the importance of mass action on eS26 binding. Finally, we include in the revised version a new toeprinting experiment, as suggested by the reviewer (Supplementary Figure 22). In this experiment, we show that preincubating ribosomes with eS26 does not change its impact on model-SD mRNA binding. Taken together, our results strongly favor the conclusion that eS26 binding to the ribosome is reversible and not biochemically stable.

All these points are now better discussed on pages 18 and 19. We clearly mention that Ss-eS26 acts rather as a factor than as a marker of specialized ribosomes. This discussion is also extended to the regulatory role of eS26 in eukaryotes. We thank the reviewer for raising this point that allowed us to improve the clarity of our conclusions.

2. The authors could describe the newly identified archaeal ribosomal proteins a little bit better.

a. The authors' analysis (e.g., Figures 3B and 4) shows that protein eS6 is conserved only in Thermoprotei, but not in other Archaea, including the closest eukaryotic relative, Asgard. How do the authors explain the origin of eS6 in eukaryotes and its absence in Asgard?

Actually, eS6 is present in Asgard but lacks domain 2. The phylogenetic distribution shown in Figure 4b concerns only domain 2 of eS6. To clarify this point, we improved the presentation of Figure 4b (also taking into account remark 2c of the referee) and added a sentence in the legend making it clear that eS6 is present in all Archaea.

b. I would expect eS6 to be labeled in Figure 1.

The protein eS6 is now labeled in Figure 1.

c. In Figure 4, could the authors also show the structures of proteins aS33, aS34 and eS6—so that a reader can judge about their size and three-dimensional structure? These are key discoveries of the paper and they are barely shown in their current figures.

We thank the reviewer for pointing this out. We now added a panel to Figure 4 (Figure 4a) showing the structures of aS33, aS34 and eS6.

d. Are the genes for proteins aS33, aS34, and eS6 organized in a single operon?

The three genes are scattered in the genome. eS6 is the first cistron of an operon containing the gene for eIF2gamma and a protein of unknown function. aS33 is the fourth cistron of an operon containing, the gene coding for a RadA homolog followed by two ORFs coding for proteins of unknown function. aS34 is the first ORF of a bicistronic operon, followed by an ORF coding for a protein of unknown function. We now mention the genetic independence of the genes coding for eS6, aS33 and aS34 in the text on Page 11, end of the first paragraph.

e. The authors should consider explicitly stating that the eS6-domain-containing protein identified in this study has the same location in the archaeal ribosome as protein eS6 in eukaryotic ribosomes. They could show this, for instance, in their Figure 3 by adding panel C.

We added a panel to Figure 3 (Figure 3c) to show the position of eS6 in the human ribosome.

Minor comments:

1. Archaeal species are known to greatly differ in the content of ribosomal proteins, often lacking one or a few proteins that are present in other archaeal species (<https://pubmed.ncbi.nlm.nih.gov/12490706/>). Because the structure of *S. solfataricus* small subunit appears to be the most complete and even includes additional proteins, I suggest the authors label all individual ribosomal proteins in the ribosome structure (instead of labeling just the archaeal-specific proteins they have identified in their study). This could help make this work an independent reference for everyone interested in the structure and evolution of the archaeal translation machinery.

We thank the reviewer for this suggestion. We now added a panel to Figure 2 (Figure 2e) showing the structure of *S. solfataricus* 30S with all ribosomal proteins labeled.

2. I would consider renaming the following sections for clarity: “Protein specificities” to “Protein features” and “Specificities of the small ribosomal subunit from *S. solfataricus*” to “Specific features of the small ribosomal subunit from *S. solfataricus*”.

These points have been corrected.

3. Consider adding essential detail when referring to the structural features of ribosomal proteins. For example, instead of writing “In addition, in eukaryotic initiation complexes, eIF1A is bound to the A site and contacts the N-terminal tail of eS30”, consider specifying the residues: “In addition, in eukaryotic initiation complexes, eIF1A is bound to the A site and contacts the N-terminal tail of eS30, including residues 2-15”. Also, make sure you use consistent naming for amino acids (e.g. currently I see “Proline 2” and “P in position 2”, which is a bit confusing, especially when you use the same “P” to refer to the P site of the ribosome).

These points have been corrected throughout the text.

To conclude, I want to restate that this work presents genuinely novel knowledge about an important biological problem. It is highly intellectually stimulating and I will be happy to support its publication in Nature Communications.

We very much appreciate the strong support of the reviewer.

Reviewer #2 (Remarks to the Author):

This paper describes a detailed study by cryo-EM and other techniques of the structure of translation initiation complexes of the crenarchaeon *Saccharolobus* (formerly *Sulfolobus*) *solfataricus*. The ribosomes of this organism have previously been shown to share structural and functional features with those of eukaryota. Many studies have been focused on the translational initiation step, which differs markedly between bacteria on one side, and eukaryotes and archaea on the other. Elucidating the details of translation initiation in archaea, especially those belonging to the TACK branch which is the closest to eukaryotes, is very important in order to understand the evolutionary history of translation, and also to unravel the relationship between archaea and eukaryotes.

The most notable results of the present study are the detailed description of translation initiation complexes with both leaderless and leadered mRNAs, including the elucidating of the role of the SD sequences (where present) and of the terminal triphosphate group in leaderless messages. Particularly notable is the discovery of two novel ribosomal proteins having a role in these processes, and the detailed elucidation of the position and function of three proteins shared by the TACK archaea and the eukaryotes. The work is well done, well conceived and technically sound. It is so rich in valuable information that it is sure to become a main reference paper for those involved in the fields of translation and evolution of translation. Among many novel and intriguing findings, I found especially interesting the description of alternative conformational states for helix 44 in the small ribosomal subunit (which may finally help to explain the long-observed and still unexplained instability of monomeric ribosomes in *S. solfataricus* and similar crenarchaea). Moreover, the paper is very well and clearly written, the methodology information is exhaustive, and the bibliography more than adequate.

We would like to thank the reviewer for these kind and positive comments.

Reviewer #3 (Remarks to the Author):

The manuscript by Bourgeois et al. reports on the functional and structural analysis of archaeal ribosomes and their complexes with either leaderless mRNA or mRNAs comprising a Shine-Dalgarno (SD) sequence. The authors report on a number of interesting novel findings regarding the specific composition and structure of the 30S ribosomal subunit of one particular archaeal species, *S. solfataricus*, which considering recent / ongoing phylogenetic studies is an interesting approach to also understand general evolutionary aspects (which are often based on ribosomal RNA as ribosomes are fundamental cellular enzymes). Two new archaea-specific ribosomal proteins were discovered (aS33 and aS34, validated by mass spectrometry and visible in the cryo electron microscopy structures), in addition to archaea-specific versions of eS6, eS26 and other proteins. Ribosomal protein aS33 and aS34 appear to co-occur in archaeal sub-species, giving insights into co-evolution. Ribosomal protein eS26 is shown to compete with mRNA and may serve as a trigger to switch between SD-leadered and leaderless mRNA 30S translation initiation complexes. Leaderless mRNA is found to comprise a 5'-triphosphate moiety that contributes to stabilization of 30S/mRNA complexes as observed in bacterial ribosomes, which is corroborated by a comparison of tri-, de- and mono-phosphorylated versions of the mRNA using toeprinting analysis. There are also some interesting features on chemical modifications of the rRNA, which compared to bacterial and eukaryotic/human ribosomes is interesting to compare with. The 3' end of the 16S rRNA is shown to be processed, which is a new finding too regarding 16S maturation in archaea. Taken together, this work puts together many complementary tools combining functional and structural analysis, which provides many novel insights into archaea-specific ribosomes and the molecular mechanism of translation initiation. The manuscript is well written and is complemented by a series of supplementary data. Some detailed points and minor corrections are suggested below for a final (minor) revision.

We would like to thank the reviewer for his support and for his positive comments.

Detailed points:

- abstract: maybe add as a conclusion (to clarify) that eS26 favors leaderless mRNA binding
Following the reviewer's suggestion, we clarified the last sentence of the abstract.

- introduction: "no bacterial-type proteins are present in the archaeal ribosome": maybe reformulate because the universal ribosomal proteins are also bacterial
We agree that this sentence was confusing. It has been deleted for greater clarity.

- introduce SD abbreviation
This has been done (Page 4).

- add "helix" to h44 to say "helix h44" (in several places); same for helix h16
This has been done throughout the text.

- which implication could have the 2 conformations of helix h44? Is there any factor-dependence on this? Or is this simply a feature reminiscent of the absence of the 50S ribosomal subunit?

Comparison of the structures with h44-up or -down revealed no change beyond the position of the distal part of helix h44. We therefore believe that the h44 helix is intrinsically mobile and that it is stabilized upon binding to 50S. This point is discussed in the text on page 7.

- usage of high Mg acetate concentration: Mg²⁺ is known to stabilize RNA complexes, but may also lead to specificity loss in interactions

We agree with the reviewer on the importance of Mg ion concentration. This concentration had been previously optimized by Londei et al, as cited in the text. We also tested several Mg concentrations in our toeprinting experiments and concluded that the optimal Mg concentration was indeed 18 mM in our conditions.

- cryo-EM processing: classifications using masks refer to focused classifications (and refinements), see a recent paper in JSB and references therein.

This has been corrected throughout the text and in Supplementary Figure 5. The recent paper in JSB is now cited.

- chemical modifications of the rRNA: could be compared with other bacterial and eukaryotic species, including human (for the few conserved across species; see for example a recent analysis in NSMB 2024, Suppl Data Table which comprises human and other eukaryotic species)

- m1acp3PSU modification found in eukaryotes, indeed e.g. in human: see PDB IDs 6QZP and 8QOI (and corresponding references)

Supplementary Table 3 has been updated with data from the NSMB article on human ribosome, as suggested by the reviewer.

- chemical modification of A1475/1476: usually this site has two dimethyl modifications (2 methyl's on each adenine), is there only one modified here? see also Suppl. Fig. 9, which seems to show only one modification, as suggested also from primer extension analysis (Suppl. Fig. 10); both in bacteria and human etc. this is a double-modification site

We agree that this site is usually doubly dimethylated, as observed in bacteria and human. Here, several lines of evidence support that, among the two N(6),N(6)-dimethyl-adenosines expected at positions 1475 and 1476, only A1475 is methylated (RT experiments and cryo-EM maps). This is discussed in the manuscript in light of the available data (Page 9, first paragraph).

- comparisons with human ribosomes: it could be useful to use the latest high-resolution work for this (NSMB 2024); in particular, the interactions described for eS26, eS25, eS30 etc. are visible in there also: eS26 is well visible and is in part close to the mRNA channel; eS25 is there and its N-ter becomes ordered upon tRNA presence; eS30 N-ter visible from 5. residue onwards.

The new panel of Figure 3 (Figure 3c) was drawn from the NSMB 2024 structure. Reference to this article is now made at different places in the manuscript.

- proposed π -stacking between Proline and glycosidic bond: from a chemistry point of view there is no aromatic system involved, hence this sounds more like a van der Waals contact; to be checked

We thank the reviewer for this remark. The sentence has been corrected accordingly (last sentence of page 9).

- 2 N-terminal glycine residues of eS25: what are the specific molecular interactions of these residues, backbone hydrogen bonding possibly? Maybe add figure panel in Suppl. Fig. 12 to show the detailed interactions

Possible hydrogen bonds involving the two terminal glycines of eS25 are now drawn in the close-up view of Figure 5A, as suggested by the reviewer. The interactions are described in the legend.

- page 11: clarify which protein gets which name (aS34 and aS33)

We agree with the reviewer that this point was not clear in the first version. This has been corrected and the Uniprot code is now mentioned with the name of the protein (Page 11, first paragraph).

- co-occurrence of ribosomal protein aS33 and aS34: is there any evidence for co-evolution, also with eS6, from sequence alignments? But on the other side they seem to be far from each other on the structure. Would AlphaFold be able to predict these 2 proteins?

Sequence alignments did not show any apparent evidence for co-evolution. However, we agree with the reviewer that the three proteins are distant in the structure. The AlphaFold predictions are consistent with the cryo-EM structures. aS33 constructed from the cryo-EM map is superimposable on an AlphaFold prediction with an rmsd of 0.58 Å for 57 C-alpha atoms compared. aS34 constructed from the cryo-EM map is superimposable on an AlphaFold model with an rmsd of 0.53 Å for 54 C-alpha atoms compared. eS6-domain 2 constructed from the cryo-EM map is superimposable on an AlphaFold model with an rmsd of 0.46 Å for 100 C-alpha atoms compared. This is now indicated in the legends of Supplementary Figures 8, 13, 14.

- page 12: stabilization of codon-anticodon duplex with uS19: has been seen also in human, see Bhaskar et al., 2020.

We were aware of this very interesting work. However, the presentation on page 12 focuses on the three uS9, uS13 and uS19 in archaeal cases and at the translation initiation step. Therefore, we preferred not to mention this eukaryotic system, which studies uS19 at the elongation step, at this point.

- “potential map”: cryo-EM maps are indeed Coulomb potential or electrostatic potential maps, but this seems to add little to the discussion in the text (unless there would be charge effects to be considered); maybe simply refer to “cryo-EM maps”

This has been corrected throughout the text.

- binding of 2 mRNA molecules: a) if so, what could this involve functionally? b) to be on the same side, are these 2 molecules clearly identified individually from the sequence visible in the cryo-EM map, i.e. are the same sequence patches observed twice (e.g. no overlap of 2 fragments visible)?

The binding site of the second mRNA molecule is illustrated in Supplementary Figure 18 with the sequence drawn together with the cryo-EM map. An important point for sequence assignment is the occurrence of the A bulge, necessary for pairing with the anti-SD sequence. This bulge is clearly visible in the cryo-EM map. The occurrence of this second mRNA molecule bound to the exit channel is likely to be due to the mass action and the short size (15 bases) of the mRNA. It would not sterically be possible with a full length mRNA.

- page 16, position of eS26 in yeast and human: also in 6QZP and 8QOI (and corresponding references); also in Fig. 5B legend; same for eS30, also in Suppl. Fig. 11A and 12B legends; in general, references should better be added to PDB IDs

This has been corrected.

- references to resolution estimation from Fourier shell correlation (FSC) should normally comprise 1) van Heel M., Keegstra W., Schutter W. G. & van Bruggen E. F. J. Arthropod hemocyanin studied by image analysis. Life Chem Rep Suppl 69–73 (1982). 2) Saxton, W. O. & Baumeister, W. The correlation averaging of a regularly arranged bacterial cell envelope protein. J. Microsc. 127, 127–138 (1982). 3) Rosenthal, P. B. & Henderson, R. Optimal determination of particle orientation, absolute hand, and contrast loss in single-particle electron cryomicroscopy. J. Mol. Biol. 333, 721–745 (2003).

The references have been added in the legend of Supplementary Figure 5.

- Suppl. Fig. 7, update of secondary structures according to the experimental structure: is it maybe possible to feedback this information into the database to have correct annotations for everyone using the database?

We thank the reviewer for this remark. The secondary structure files were corrected according to the experimental structures. We will send an update to <https://crw-site.chemistry.gatech.edu/>.

- it could be useful to have a validation report for the refined coordinates & pdb deposition, for example to check for clashes in the atomic model etc.

The validation reports, the coordinate files and the cryo-EM maps were too big to be deposited on the submission site. However, we deposited all these data on the CNRS cloud and gave the link to the reviewers through a supplementary file named SI-link-to-PDB-and-Cryo-EM-Maps. We apologize if our procedure was not clear enough. Here is the link to the files <https://mycore.core-cloud.net/index.php/s/oTcpxuomp6QX8FB>. This link will also be available in the same SI-link-to-PDB-and-Cryo-EM-Maps supplementary file for reviewers. Moreover, validation reports and PDB files are deposited on the submission website and available as a Figshare private link.

- generic point: cryo-EM maps, half maps, masks etc. and fully refined atomic models should be deposited in the EMDB and PDB; deposition of representative data onto the EMPIAR data base could be considered also

All the files EMDB, PDB, SM had been deposited in appropriate databases. Deposition numbers are indicated at the end of the manuscript.

- figures look good; suggestion for Fig. 8: maybe transfer to Suppl. Figs?
Thank you for the opportunity to review this work.

We have carefully considered this suggestion. To keep some visibility to this information, we would prefer it to appear in a main Figure. However, we reduced the size of the figure to a single column.

Bruno Klaholz